# Diversity of trypanosomes in humans and cattle in the HAT foci Mandoul and Maro, Southern Chad—A matter of concern for zoonotic potential?

**Mahamat Alhadj Moussa Ibrahim**[1,2]*, **Judith Sophie Weber**[1,3], **Sen Claudine Henriette Ngomtcho**[1,4], **Djoukzoumka Signaboubo**[1], **Petra Berger**[1], **Hassane Mahamat Hassane**[5], **Sørge Kelm**[1]*

**1** Centre for Biomolecular Interactions Bremen, Department of Biology and Chemistry, University of Bremen, Bremen, Germany, **2** Department of Biology, Faculty of Exacts and Applied sciences, University of N'Djamena, N'Djamena, Chad, **3** Centre for Marine Environmental Sciences MARUM, University of Bremen, Bremen, Germany, **4** Department of Microbiology, Haematology and Immunology, Faculty of medicine and pharmaceutical Sciences, University of Dschang, Dschang, Cameroon, **5** Institut de Recherche en Elevage pour le Développement, N'Djamena, Chad

* ibrahimmahamat21@gmail.com (MAMI); skelm@uni-bremen.de (SK)

**Data Availability Statement:** All relevant data can be found in the paper and its supporting information files.

## Abstract

### Background

African trypanosomes are parasites mainly transmitted by tsetse flies. They cause trypano-somiasis in humans (HAT) and animals (AAT). In Chad, HAT/AAT are endemic. This study investigates the diversity and distribution of trypanosomes in Mandoul, an isolated area where a tsetse control campaign is ongoing, and Maro, an area bordering the Central African Republic (CAR) where the control had not started.

### Methods

717 human and 540 cattle blood samples were collected, and 177 tsetse flies were caught. Trypanosomal DNA was detected using PCR targeting internal transcribed spacer 1 (ITS1) and glycosomal glyceraldehyde-3 phosphate dehydrogenase (*gGAPDH*), followed by amplicon sequencing.

### Results

Trypanosomal DNA was identified in 14 human samples, 227 cattle samples, and in tsetse. Besides *T. b. gambiense*, *T. congolense* was detected in human in Maro. In Mandoul, DNA from an unknown *Trypanosoma sp.*-129-H was detected in a human with a history of a cured HAT infection and persisting symptoms. In cattle and tsetse samples from Maro, *T. godfreyi* and *T. grayi* were detected besides the known animal pathogens, in addition to *T. theileri* (in cattle) and *T. simiae* (in tsetse). Furthermore, in Maro, evidence for additional unknown trypanosomes was obtained in tsetse. In contrast, in the Mandoul area, only *T.*

**Funding:** This study's financial support came from the Islamic Development Bank (IsDB: https://www.isdb.org/) providing stipend and bench fees to IMAM _36/11210149; 600029774. Deutscher Akademischer Austauschdienst (DAAD: https://www.daad.de/de/), allowing sandwich program stipend to SCHN, CN_A/12/97080). Deutsche Forschungsgemeinschaft (DFG: https://www.dfg.de/), project grants to SK, Ke428/10-1 and Ke428/13-1). The funders had no role in study design, data collection and analysis, decision to publish, or prepa-ration of the manuscript.

**Competing interests:** The authors have declared that no competing interests exist.

*theileri*, *T. simiae*, and *T. vivax* DNA was identified in cattle. Genetic diversity was most prominent in *T. vivax* and *T. theileri*.

## Conclusion

Tsetse control activities in Mandoul reduced the tsetse population and thus the pathogenic parasites. Nevertheless, *T. theileri*, *T. vivax*, and *T. simiae* are frequent in cattle suggesting transmission by other insect vectors. In contrast, in Maro, transhumance to/from Central African Republic and no tsetse control may have led to the high diversity and frequency of trypanosomes observed including HAT/AAT pathogenic species. Active HAT infections stress the need to enforce monitoring and control campaigns. Additionally, the diverse trypanosome species in humans and cattle indicate the necessity to investigate the infectivity of the unknown trypanosomes regarding their zoonotic potential. Finally, this study should be widened to other trypanosome hosts to capture the whole diversity of circulating trypanosomes.

## Author summary

Sleeping sickness (HAT) is a public health problem in 36 African countries. In Chad, 5 active foci are present in the Southern part. It is caused by trypanosomes, parasites causing disease in humans and livestock. Tsetse flies, the vectors of trypanosomes, declined in the Mandoul focus due to the impact of vector control coupled with active/passive screening and treatment campaigns. In the Maro focus, where such campaigns were absent during these surveys, HAT cases were reported recently. We carried out a study on circulating trypanosomes in humans, cattle and tsetse in these two foci. The results confirmed a reduction of the tsetse population and pathogenic trypanosomes of human and cattle in Mandoul. However, an unknown trypanosome was identified in a human and high frequency of *T. theileri* (known as non-pathogenic) was found in cattle. In contrast, in Maro, a high diversity of trypanosomes was observed, including *T. b. gambiense* and *T. congolense* in humans and several unknown trypanosomes in tsetse. These observations provide evidence of the circulating trypanosomes in the area that recommend widening the investigation to other mammalian hosts and mechanical vectors and considering and monitoring a possible zoonotic potential with the unknown trypanosome and *T. congolense* in humans.

## Introduction

Human African Trypanosomiasis (HAT), known as Sleeping Sickness, and Animal African Trypanosomiasis (AAT), known as Nagana, are vector born parasitic diseases of humans and livestock caused by the transmission of extracellular protozoans of the genus *Trypanosoma*. In Central and West Africa, HAT is caused by *T. brucei gambiense*, leading to the chronic form, whereas in East and South Africa, it is caused by *T. b. rhodesiense*, leading to the acute form [1,2]. Millions of people in 36 sub-Saharan African countries are at different levels of risk of infection [3], and WHO had the final goal of sustainable HAT elimination (zero cases) by 2030. AAT occurs in ruminants, camels, equines, swine, and carnivores. The endemic disease severely reduces livestock productivity, and thus also the wealth of livestock farmers and the

nutritional well-being of the entire population [4]. The disease is widely distributed across the tsetse-infested belt of the African continent, covering about 10 million km$^2$. In this belt, approximately 60 million cattle are at risk of infection [5]. Otherwise, this tsetse-infested belt is known to be fertile land, well suited for agriculture and livestock production in Africa [6,7].

Based on their transmission paths, trypanosomes are divided into Salivaria and Stercoraria. Salivaria are transmitted in the saliva of the vector as it feeds on host blood, whereas Stercoraria are transmitted through vector feces. Among the Salivaria, *T. vivax*, *T. congolense* and *T. b. brucei* are the three most important species pathogenic for livestock and responsible for considerable production losses and morbidity [8]. Three subgenera have been defined in Salivaria trypanosomes: *T. vivax* belongs to the subgenus *Duttonella*; *T. congolense*, *T. simiae* and *T. godfreyi* [9] to *Nannomonas*; and *T. brucei* (with the sub-species *T. b. brucei*, *T. b. gambiense* and *T. b. rhodesiense*), *T. evansi* and *T. equiperdum* to *Trypanozoon* [10,11]. Members of the Stercoraria are the South American *T. cruzi*, the cosmopolitan *T. melophagium* and *T. theileri* [12] and the African *T. grayi* [13]. Among the Stercoraria, *T. theileri* is a parasite of cattle with global distribution, often occurring with high incidence [14].

The main insect vectors of African trypanosomes are tsetse flies of the genus *Glossina* (Glossinidae: Diptera). However, the parasites can also be transmitted mechanically by other biting flies such as tabanids and *Stomoxys* [15,16]. Other trypanosomes, as *T. theileri*, are mainly transmitted by tabanids.

Chad is part of the trypanosomes endemic zone, with about 65 000 km$^2$ in the southern part of the country being infested with tsetse flies [17]. However, the extension of the infested area is uncertain due to the lack of reliable recent survey data. In the endemic zone, agriculture activities are extensively practised, and after the rainy season, pastoralists looking for grass and crops residues for their livestock enter the area, often for more than 6 months. The general livestock census carried out in 2015 showed that Chad had more than 93 million cattle, sheep, goats, camels and equines [18]. Like agriculture, the livestock sector is one of the main contributors to the economy of the country [19]. However, AAT has remained a major obstacle to the development of this sector, which employs more than 40% of the population [18]. Chad also faces the public health problem HAT. This is currently present in 5 historical HAT foci Moïssala, Tapol, Goré, Mandoul, and Maro [20]. Mandoul and Maro are the most known active foci, and there are still new cases notified [21]. The Programme National de Lutte contre la Trypanosomiase Humaine Africaine (PNLTHA) and l'Institut de Recherche en Élevage pour le Développement (IRED) with their partners such as the Foundation for Innovative New Diagnostics (FIND), World Health Organisation (WHO), Institut de Recherche pour le Développement (IRD), and Liverpool School of Tropical Medicine (LSTM) are monitoring the disease in Mandoul and most recently in Maro. They are applying tsetse control, and human screenings for *T. b. gambiense* and treatment campaigns to reduce HAT infection risks. This includes usage of Tiny Targets [22], small blue-coloured panels of cloths attracting tsetse, impregnated with insecticide and deployed along river banks where tsetse flies concentrate [23]. The HAT surveillance and tsetse control had started in the Mandoul focus in 2014 and are ongoing. The strategies effectively reduced the tsetse fly populations and the HAT cases [22]. In contrast, in Maro, the campaigns have not started when this study was undertaken, and there was a resurgence of new cases.

Animals may harbour the human pathogenic species, serving as a reservoir [24]. On the other hand, infections in humans with animal-pathogenic trypanosomes can occur in rare cases [25]. About 19 cases of atypical human trypanosomes (a-HT) [26], among them *T. b. brucei*, *T. congolense*, *T. vivax*, and *T. evansi* which are considered non-infective to humans, have been reported.

We aimed to investigate the circulating trypanosomes, including the occurrence of potentially zoonotic species in humans and livestock, and in their biological tsetse vector in two active HAT foci, Mandoul and Maro. Mandoul is an area with ongoing HAT surveillances and tsetse control operation, while no such activities were carried out at the time of the surveys in Maro. Taking advantage of the widely used molecular techniques, PCR-based methods targeting trypanosomal internal transcribed spacer I (ITS1) region [27,28] and glycosomal glyceraldehyde-3-phosphate dehydrogenase (*gGAPDH*) gene combined with sequencing [29,30], were used to identify trypanosome species in humans and cattle blood samples and tsetse fly tissues. In a time of ongoing tsetse control in Chad to reduce the risk of HAT and AAT infections, the study will contribute to the monitoring strategies, by providing the genetic diversity of circulating trypanosomes, on the way to achieve the goal of diseases elimination.

## Methods

### Ethics statement

This study conducted on the distribution of trypanosomes in human, cattle and tsetse flies in Southern Chad was approved in December 2016 by the national bioethics committee under the number 585/PR/PM/MESRI/SEESRI/SG/2016. Detailed protocol and consent documents were submitted to the committee as well as wide information concerning the purpose of the study, provided to the targeted populations. Written consent was obtained from all participants, including those from parents of children under 18 years old.

### Study areas

The Mandoul and Maro HAT foci are located in Southern Chad (Fig 1, S1 Text for details). As for the tsetse fly habitat, Mandoul represents an area where flies are restricted to the swamps formed at the southern limit of the Mandoul river. As the river flows northwards, the swamp deteriorates into a marshy habitat, unsuitable for tsetse. As a result, the population is isolated. Vector control operations with the annual deployment of Tiny Targets started in 2014 [22]. However, Maro is located in far Southern Chad (Fig 1). Many rivers and their multiple tributaries cross the focus; the most important is the Chari River and its confluent, the Grand Sido, which mark the border with CAR. Tsetse habitat is configured by the thin riverine vegetation along the banks of the rivers. Vector control operations started on the Chadian bank in 2018 [21], with annual deployments of Tiny Targets, after conducted most of these surveys. No similar operations have been implemented across the border.

### Human surveys

Surveys were conducted in February 2017, March, and June 2018. In each surveyed area, eight randomly selected villages were visited and a military camp included at the request of its inhabitants. In order to proceed with the selection, households were numbered and drawn for participation. A chosen household included all its members automatically. The number of households and participants surveyed per village depended on its population and the individuals who consent for participation. However, we collected no blood from children under five years old.

The open-source Epidemiologic statistics for public health software Version 3.01 (http://www.openepi.com/SampleSize/SSPropor.htm) was used to estimate the human sample size that should be included in the survey. Based on the estimated population recorded from the institutions in charge of HAT control in Chad which were published later, the Mandoul focus includes 114 human settlements with 38,674 inhabitants [22]. In comparison, Maro had 45

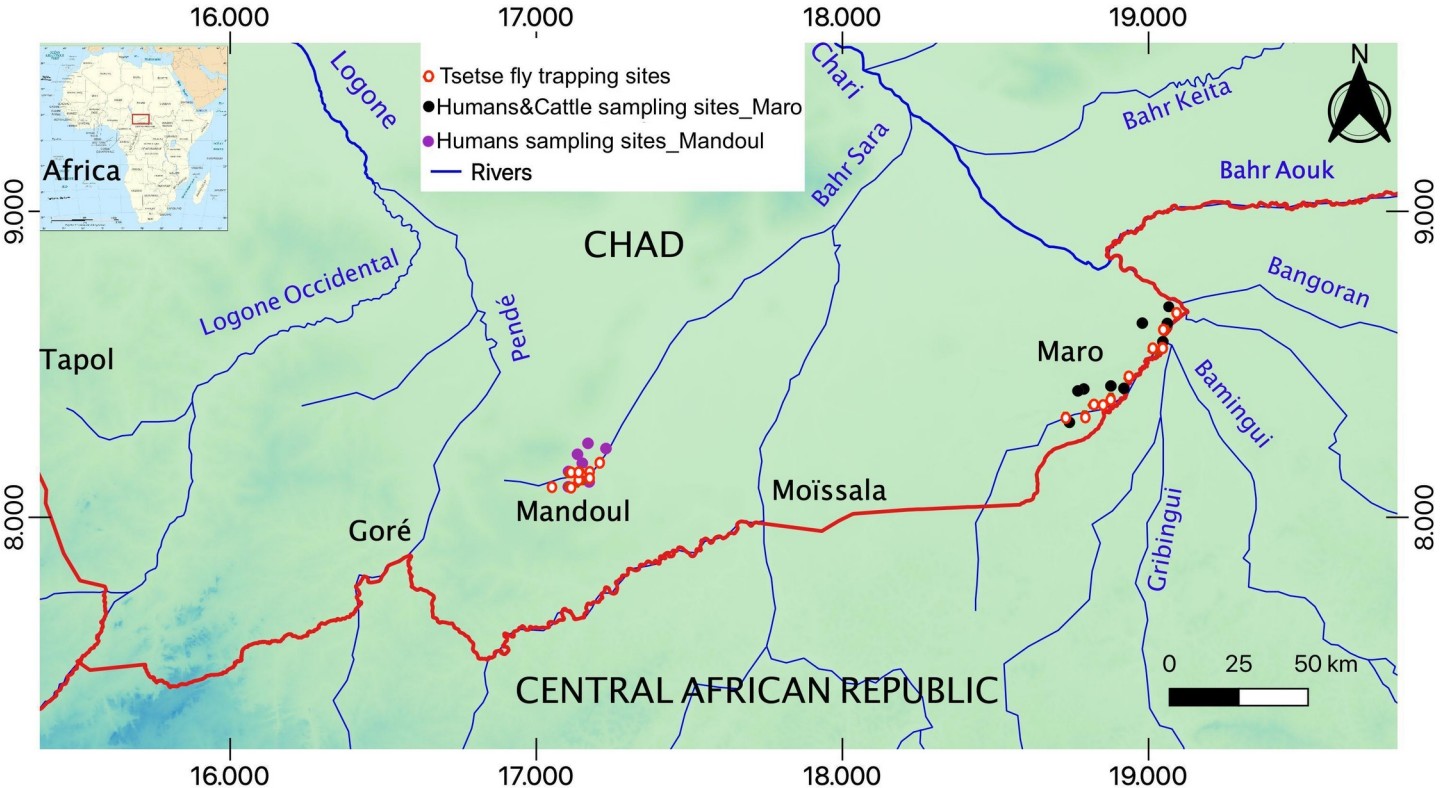

**Fig 1. Map showing humans and cattle sampling sites and tsetse trapping spots in the Mandoul and the Maro foci in Southern Chad; generated from SRTM (Shuttle Radar Topography) topo 30 data, easily accessible and free to download from the site: https://earthexplorer.usgs.gov/.**

settlements with 14,532 inhabitants in 2017 [21]. The present study used these numbers as the total populations from which the sample sizes were calculated. With an accepted margin of error of 5% and 95% confidence interval, the sample sizes required were 381 in the Mandoul and 375 in the Maro.

### Cattle surveys

We surveyed the cattle in January, March, June, and November 2018. Sedentary villages, semi-nomadic camps, a nomadic settlement, and a refugee camp were included. Six out of the nine villages selected were the same as those included in the human survey. Representative cattle, from each herd, randomly chosen, were included in this study. Though we strived from random selection, the animals were partly chosen by the herdsmen themselves, presenting animals exhibiting symptoms rather than healthy animals. Similar to that of humans, the open-source Epidemiologic statistics for public health Version 3.01 was used to estimate the sample size of the study. Mandoul has about 14,000 cattle [31], while Maro has over 55,000 [32]. With an accepted margin of error of 5% and 95% confidence interval, the sample sizes required were 374 for Mandoul and 382 for Maro.

The questionnaires were filled during the survey. They addressed the number of animals in the herd, the breeding system, breeds, source of water and nutritional support, health status including symptoms, morbidity and mortality, animal vaccinal status and drug usage, sex, age of the animals, and the herdsmen's education level. Each herdsman answered questionnaires with the support of a local translator on the same day of blood collection.

## Human and cattle blood collection and processing

About 5 to 7 mL of blood was collected from the radial vein (venipuncture) of each human participant using vacutainer butterfly needles. 7 to 10 mL was taken from the jugular vein of each animal using a syringe. Collected blood was then directly transferred/connected into a labelled blood collection tube (or vacutainer tube) containing EDTA. The tubes were processed and the pack cell volumes (PCVs) measured. 200 μL of whole blood were pipetted into 1.5 mL labelled cryotube, and 50 μL were added to 150 μL of Nucleic Acid Preservative Agent, NAPA (25 mM sodium citrate, 10 mM EDTA, 5.3 M ammonium sulfate, pH 7.5), in a separate cryotube.

## Tsetse fly collections and processing

Entomological surveys were conducted in February 2017, March, June, and November 2018. Biconical traps were used for this study. Trapped tsetse were daily collected and dissected as detailed in S2 Text. Proboscises were collected from all flies and stored in 200 μL NAPA. The guts were dissected and kept separately in a 1.5 mL labelled cryotube from live flies [28,29]. The remaining body of dead tsetse flies (TRB) were kept in 500 μL ethanol after removing the proboscis. Species were morphologically identified and molecularly confirmed by PCR and sequencing, following the procedures described by Shaida *et al.* [33].

## DNA extraction and quantification

DNeasy Blood and Tissue Kit (Qiagen, Hilden, Germany) was used to purify DNA from human and cattle blood and tsetse homogenised gut according to the manufacturer instructions with slight modification using 100 μL of treated blood and 100 μL of elution buffer. Photometric quantification of extracted DNA at 260 nm wavelength was performed on a Nanodrop 1000 (Thermo Fisher Scientific, Dreieich, Germany) [28,29].

DNA from proboscis was extracted using a crude extraction method. The proboscis was incubated at 55˚C for 1 h with 55 μL 0.33 mg/mL Proteinase K (Thermo Fisher Scientific, Dreieich, Germany), diluted in 10 mM phosphate buffer, pH 7.4. Heat inactivation of the enzyme followed at 80˚C for 45 min [29].

DNA was extracted from the tsetse remaining body (TRB, without proboscis, legs, and wings) using 5% Chelex-100 Resin (BIO-RAD, Hercules, California, USA) [34]. 100 μL was applied on a slightly squashed TRB with a pestle in a 1.5 mL tube. The mixture was incubated at 56˚C for 30 min, vortexed and incubated for an additional 5 min at 95˚C. The extracted DNA was mixed thoroughly before brief centrifugation at 7000 rpm for 45 s and kept at -80˚C.

## Molecular amplification and identification of trypanosome species

Nested PCR targeting the ITS1 region of the ribosomal RNA gene locus was carried out to identify the species of trypanosomes. The gene was chosen because of its high copy number and its interspecific length variation, which had previously been used for species identification [27,34]. For this purpose, established generic and specific primers [27,28] were used (see S1 Table for details) following previously published procedures [28,29] adapted to the sample types (blood or tsetse tissues). 25 μL of a master mix containing 2 μM of each outer primer (Sigma-Aldrich, Darmstadt, Germany), 200 μM dNTPs, 2.5 Units Dream*Taq* polymerase, 1x Dream*Taq* buffer (all from Thermo Scientific) and template DNA was used. The volume was 1 μL for human, cattle, TRB, and proboscis template DNA, and 5 μL for tsetse gut tissue template DNA [28,29]. The cycling conditions for both reactions were as follows: initial denaturation at 95˚C for 3 min followed by 30 cycles at 94˚C for 60 s, 54˚C for 30 s, 72˚C for 30 s and

final elongation at 72˚C for 5 min. Trypanosome species were initially identified based on the size of their ITS1 PCR products, estimated by agarose electrophoresis (for details see S3 Text). Then, the amplicons were purified and subcloned into a linearised pJET 1.2/blunt plasmid using the CloneJET PCR Cloning Kit (Thermo Fisher Scientific) and sequenced by Sanger sequencing following the protocol detailed in S3 Text.

For confirmation of the ITS1 analysis and performing of phylogenetic analyses, a nested PCR targeting the partial *gGAPDH* gene was carried out. *gGAPDH* is a ubiquitous, essential glycolytic enzyme and has a slow rate of molecular evolution making it suitable for studying evolution over large time-scales [30] and therefore, it has been a marker of choice for phylogenetic analysis. A master mix was prepared as described for ITS-1 nested PCR except for the respective primers (S1 Table [29,30]). Reaction conditions, for the first PCR, were 95˚C for 3 min, followed by 30 cycles at 95˚C for 1 min, 55˚C for 30 s, and a final extension at 72˚C for 10 min. Similar conditions as in the first reaction were used in the second, except the annealing temperature changed to 52˚C. PCR products were separated by gel electrophoresis as described above. Amplicons were purified and sequenced as detailed in S3 Text.

## Sequences read and phylogenetic tree construction

Sequences were read, analysed and aligned using Geneious Pro 5.5.9 [35]. Alignments of ITS1 and *gGAPDH* sequences were done with Gap open penalty 15 and Gap extension penalty 5. The sequences were aligned against the GenBank database using nucleotide BLAST from NCBI and TritrypDB.

MEGAX software was used to investigate the phylogenetic relationship of trypanosomes based on their *gGAPDH* sequences [36]. The DNA sequences were imported from Geneious and aligned with the MUSCLE algorithm method together with the reference sequences retrieved from the GenBank database. The evolutionary history was inferred using the Neighbour-Joining method. The evolutionary distances were computed with the Maximum Composite Likelihood method [37].

## Statistical analysis

The frequencies (in percentage) of trypanosome species in humans, cattle, and tsetse fly samples, were obtained using the Clopper-Pearson binomial test with the lower and upper limits of the 95% confidence interval. Pearson Chi-Square tests were applied to compare, in these sampled animals, trypanosomes frequency to age groups, collection periods and cattle breed, as well as trypanosomes frequency to nomadic, sedentary and refugees' cattle. Student's t-test (unpaired, two-tailed) was used to compare the mean PCV values of recorded healthy and sick cattle on the one hand, and the mean PCV values of trypanosome PCR positive and negative cattle, on the other. In these collected samples, differences were tested for significance at $p<0.05$ using SPSS v.22.0 (IBM, USA). Prism version 7.0a was used for constructing the graphs and Microsoft Excel for managing raw data.

## Results

### Human survey data

A total of 889 human participants were recruited during the surveys to cover the estimated sample sizes. 409 were from the Mandoul sleeping sickness focus and 480 from the Maro focus. Among those, a total of 717 agreed to give blood samples. 306 samples were collected from the Mandoul focus, less than the study design (381 participants), and 411 from the Maro focus including 19 individuals of a military camp, fulfilling the study deign (375 participants)

(see S2 Table for details). Recruitment of participants was carried out in 13 sedentary villages (553; 77.13%), 2 semi-nomadic camps (121; 16.87%), 1 nomadic colony (24; 3.35%), and 1 military camp (19; 2.65%), at the rate of 8 settlements in Mandoul and 9 in Maro. 19 to 70 blood samples were collected from each of those villages depending on the size of the population and agreement of the participants. Both genders were represented, 371 males (52%), 340 females (47%) and 6 (1%) with unrecorded sex. Children under 5 years were excluded from blood collection. The main activities practised by the participants are agriculture, livestock, and fisheries.

### Trypanosomes identification in humans

Analysis of *Trypanosoma-gGAPDH* gene performed on DNA extracted from human blood samples revealed *T. b. gambiense*, *T. congolense* and *Trypanosoma sp.*-129-H. 14 human samples were positive, giving an overall trypanosome frequency of 2.0% (95% CI: 1.1–3.3%) in the two foci. Strikingly, in the Maro focus, *T. b. gambiense* DNA was identified in two samples 0.5% (95% CI: 0.1–1.8%) and *T. congolense* in 11 leading to an observed frequency of 2.8% (95% CI: 1.4–5.0%) (Fig 2). In Maro, the trypanosome's frequency was 3.3% (95% CI: 1.8–5.6%) and none of the military participants was trypanosome DNA positive. In the Mandoul samples, there was no evidence for trypanosomes DNA in humans apart from an unknown trypanosome termed *Trypanosoma sp.*-129-H found in one individual (0.3% (95% CI: 0.0–1.8%)). All obtained sequences (see S1 Appendix for details) were aligned against the GenBank database using nucleotide BLAST in NCBI and TritrypDB. *T. congolense* and *T. b. gambi*ense sequences were between 99.4% to 100% similar to described sequences in GenBank, except one *T. congolense* sample which showed only 87% similarity, which was later on confirmed by *T. congolense* specific PCR. The sequence of the *Trypanosoma sp.*-129-H was 84% similar to *Trypanosomatidae sp.* LW-2010b (Accession number HQ263665).

Referring to Adams *et al.*, [27] and Ngomtcho *et al.*, [28] concerning *Trypanosoma* species-specific amplicon sizes, all human samples were screened for *Trypanosoma* DNA targeting ITS1 region and were later on confirmed by sequencing. The analyses revealed 1 positive sample; 0.3% (95% CI: 0.0–1.8%) frequency. The sequenced amplicon resulted in 372 base pairs

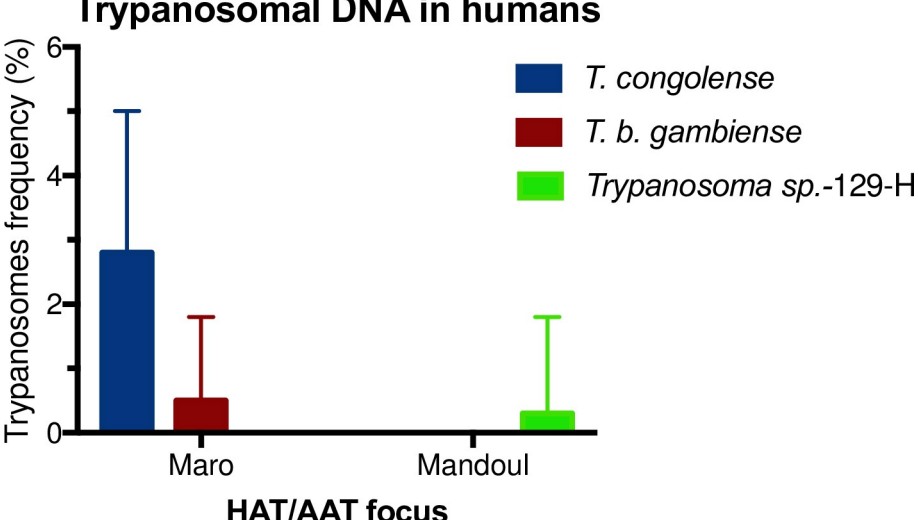

**Fig 2. Trypanosomes frequency in humans in the Mandoul and Maro HAT/AAT foci.** Error bars represent the upper limit of the 95% CI.

(see S1 Appendix and S1 Fig; *Trypanosoma sp.*-129-H). The sequence showed 97% similarity to *Trypanosomatida sp.*, JN673399 previously detected in a hyaena in Tanzania. Interestingly, this is the only sample identified by ITS1-PCR in humans at the Mandoul focus. Furthermore, the presence of trypanosomal DNA was confirmed when targeting *gGAPDH* gene, which was 84% identical to *Trypanosomatidae sp*. HQ263665, detected in *Drosophila obscura*.

Overall, 69% (9) of the cases positive for trypanosomal DNA in the Maro focus were individuals younger than 20 years. Furthermore, most of them (7) were between 10 and 15 years old. This cannot be explained only by a higher percentage of younger participants, as only 42.6% of the participants were under 20 years old, and 14.11% were between 10 and 15 years.

## Cattle survey data

A total of 540 cattle blood samples were collected from 93 herds. 462 cattle were from Maro and 78 from Mandoul. The number of cattle sampled in Mandoul is low due to an unexpected anthrax outbreak during the planned survey in the area. Surveys were carried out in 5 sedentary villages, 2 semi-nomadic camps, 1 nomadic settlement, and 1 refugee camp (S3 Table). 61.2% (95% CI: 56.9–65.4%) of blood samples were collected from females, and 38.8% (95% CI: 34.6–43.1%) from males. Regarding the breeds, 87.6% (95% CI: 84.5–90.3%) were Arab zebu, 6.9% (95% CI: 4.9–9.4%) White Fulani, and 5.5% (95% CI: 3.7–7.9%) were M'bororo breed. 3% of the cattle have had sex and breed unrecorded. The semi-nomadic and nomadic breeders of these surveys owned exclusively Arab zebu breed from whom 275 cattle were sampled, while the sedentary villagers were raising 1, 2 and/or 3 breeds from whom 265 were sampled. 88% (95% CI: 85.5–93.1%) of the sedentary groups were breeding cattle for traction and farming work, while 99% (95% CI: 98.8–100%) of the nomadic and semi-nomadic groups were practising an extensive livestock system. The reason for the transhumance was lack of grasses in the areas during the dry season.

## Trypanosomes distribution in cattle using ITS1 nested PCR

ITS1-nested PCR was performed on 540 cattle blood samples. *T. congolense*, *T. brucei ssp.*, *T. simiae*, *T. theileri*, *T. grayi*, *T. godfreyi*, and *T. vivax* were detected (see S1 Fig for amplicon sizes). Similar to a previous report from Cameroon [28], PCR products of about 150 base pairs were observed in 22 samples, but not taken into account for the frequency of trypanosomes. 223 (41.3%; 95% CI: 37.1–45.6%) cattle samples were positive for *Trypanosoma* DNA (S3 Table). In the Maro focus, 159 (34.4%; 95% CI: 30.1–38.9%) out of 462 cattle contained trypanosomal DNA, whereas in the Mandoul focus 64 (82.1%; 95% CI: 71.7–89.8%) out of 78 cattle sampled were positive.

At the Maro focus, *T. congolense* DNA was found most frequently, followed by *T. vivax*, *T. theileri*, *T. brucei ssp.*, *T. grayi*, and *T. godfreyi* (Fig 3A). In contrast, in the Mandoul area (Fig 3B), *T. theileri* was by far most frequently found (91.0% (95% CI: 81.5–96.6%) of all positive samples), and only a few animals were positive for *T. vivax* and *T. simiae*. The latter was not detected in Maro.

Mixed infections with two or three different trypanosome species, commonly observed in cattle in previous studies [38], were also found in this study (S3 Table). The most common trypanosome in co-infections was *T. vivax*. *T. congolense* in association with *T. brucei ssp*. were present in 16 cattle, followed by *T. theileri* in co-occurrence with *T. vivax* (5 cases). In 1 cow, the 3 pathogenic species (*T. congolense*, *T. brucei ssp.*, and *T. vivax*) were observed together. For details of mixed infections of other trypanosomes species, see the additional file in S3 Table.

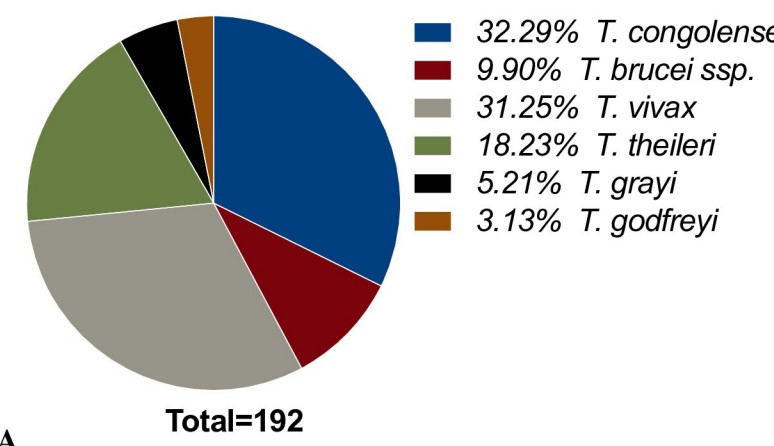

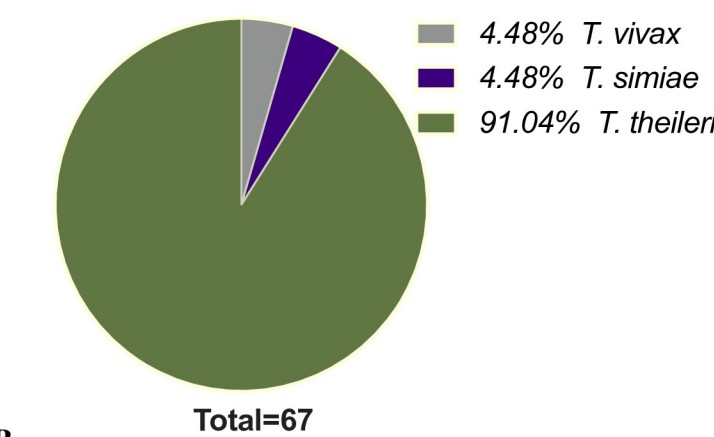

**Fig 3. Overall trypanosomes frequency in cattle.** A- Maro HAT/AAT focus; B- Mandoul HAT/AAT focus.

## Distribution of trypanosomes depending on temporal parameter, migration, age, and cattle breeds of Maro's cattle

Regarding the community structures of Maro's cattle included in the survey, the pathogenic trypanosomes were more frequent (statistically significant, $X^2$ = 11.87; p<0.05) in nomadic than in sedentary cattle (Table 1). Within the nomadic and sedentary cattle groups, *T. vivax* represented 15.6% (95% CI: 11.6–20.5%) and 10.3% (95% CI:0.6–16.0%), *T. congolense* 15.3% (95% CI: 11.2–20.1) and 12.1% (95% CI: 7.6–18.1%), and *T. brucei ssp.* 5.5% (95% CI: 3.1–8.8%) and 2.4% (95% CI: 0.7–0.61%), respectively (Fig 4A). In animals at the refugee camp, these parasites were not found. However, *T. theileri* the worldwide spread bovid trypanosome was identified almost at the same frequency (Fig 4A) at nomadic, sedentary, and refugees' sites, while *T. grayi* was identified in only few cattle samples of these groups. What stands out is that the cattle of the refugee group were significantly less infected ($X^2$ = 7.52; p<0.05) with any trypanosome species than the other sedentary cattle and nomadic group (Table 1). Also, the pathogenic species are more prominent in the nomadic group.

**Table 1. Effect of transhumance activities, collection periods, age, breeds, and sex on trypanosomes frequency in Maro.**

| | Overall trypanosomes | | | Pathogenic/Non-pathogenic trypanosomes | | |
|---|---|---|---|---|---|---|
| | Positive within group N (%) | Overall positive N (%) | $X^2$, df, p-value | Pathogenic N (%) | Non-Pathogenic N (%) | $X^2$, df, p-value |
| **Community, n = 462** | | | | | | |
| Nomadic | 106 (38.5) | 159 (34.4) | $X^2 = 7.52$, df = 2, p = 0.023 | 84 (18.2) | 22 (4.8) | $X^2 = 11.87$, df = 4, p = 0.018 |
| Sedentary | 50 (30.3) | | | 38 (8.2) | 12 (2.6) | |
| Refugee | 3 (13.6) | | | 0 (0.0) | 3 (0.6) | |
| **Collection period, n = 255** | | | | | | |
| January | 71 (39.7) | 101 (39.6) | $X^2 = 28.46$, df = 2, p = 0.000 | 60 (23.5) | 11 (4.3) | $X^2 = 42.67$, df = 4, p = 0.000 |
| March | 19 (86.4) | | | 11 (4.3) | 8 (3.1) | |
| November | 11 (20.4) | | | 10 (3.9) | 1 (0.4) | |
| **Age groups (yrs), n = 452** | | | | | | |
| < 2.5 | 24 (24.5) | 156 (34.5) | $X^2 = 7.38$, df = 3, p = 0.061 | 12 (2.6) | 12 (2.6) | $X^2 = 17.19$, df = 6, p = 0.009 |
| 2.5 to 5 | 62 (34.1) | | | 49 (10.6) | 13 (2.8) | |
| >5 | 70 (40.7) | | | 59 (12.8) | 11 (2.4) | |
| **Cattle Breed, n = 452** | | | | | | |
| Arab Zebu | 135 (34.2) | 156 (34.5) | $X^2 = 1.16$, df = 2, p = 0.559 | 106 (23.5) | 29 (6.4) | $X^2 = 3.51$, df = 4, p = 0.476 |
| White Fulani | 15 (41.7) | | | 11 (2.4) | 4 (0.9) | |
| M'bororo | 6 (28.6) | | | 3 (0.7) | 3 (0.7) | |
| **Sex, n = 452** | | | | | | |
| Male | 55 (35.3) | 156 (34.6) | $X^2 = 0.32$, df = 1, p = 0.571 | 37 (8.2) | 18 (4.0) | $X^2 = 4.68$, df = 2, p = 0.096 |
| Female | 101 (64.7) | | | 83 (18.4) | 18 (4.0) | |
| **Health status, n = 452** | | | | | | |
| Sick | 111 (71.2) | 156 (34.5) | $X^2 = 1.14$, df = 1, p = 0.285 | 91 (20.1) | 20 (4.4) | $X^2 = 6.36$, df = 2, p = 0.041 |
| Apparently-healthy | 45 (28.8) | | | 29 (6.4) | 16 (3.5) | |

Looking at the seasonal distribution of 255 samples entirely collected from two nomadic settlements in the Maro focus, all identified pathogenic trypanosome species were present either at the beginning of the dry season (November), in the middle (January) or near the end of the dry season (March) (Fig 4B). In November, the frequency of *T. congolense*, *T. brucei ssp.*, and *T. vivax* were similar. In January, however, there was an increase in the frequency of *T. congolense* (18.4% (95% CI: 13.0–24.9%)) and *T. vivax* (15.6% (95% CI: 10.7–21.8%)) and a decrease of *T. brucei ssp.* (4.5% (95% CI (1.9–8.6%)). In March, while *T. brucei ssp.* frequency from these data was the same as in January, the highest rate of *T. vivax* (36.4% (95% CI:17.2–59.3%)) and *T. theileri* (45.5% (95% CI: 24.4–67.8%)) were observed. What stands out in the overall samples was in November (Table 1), the cattle were significantly less infected ($X^2 =$ 28.4; p<0.000) with any trypanosome species than in January, and in March; with the presence of pathogenic species over all the studied period and significant increase of *T. theileri* in March.

Trypanosome-positive cattle were disseminated according to their age groups (Fig 4C). In young cattle (<2.5 years), *T. congolense*, *T. vivax*, and *T. brucei ssp.*, the pathogenic species, were at the lowest frequency compared to that in mature group (2.5 to 5 years) and elder cattle (>5 years). The few cases of *T. grayi* and *T. godfreyi* observed were distributed in all age groups. In summary (Table 1), young cattle were significantly less trypanosomal DNA positive

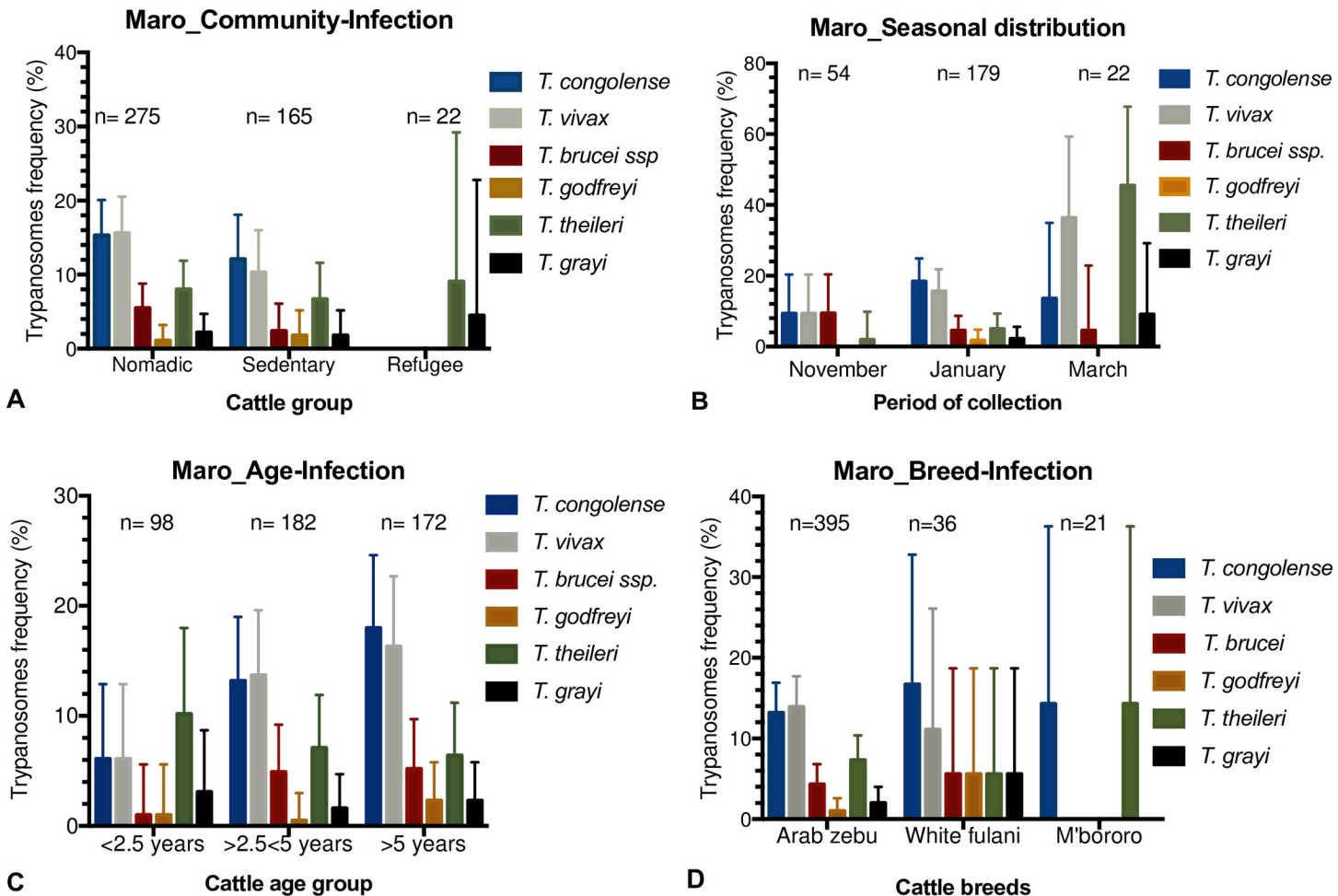

**Fig 4. Distribution of trypanosomes depending on seasonal aspect, migration, age and cattle breeds.** n = number of animals included. **A**- Migration-Infection; **B**-Seasonal distribution in 2018; **C**- Age-Infection; **D**- Breed-Infection. Error bars represent the upper limit of the 95% CI.

($X^2 = 7.38$; $p < 0.05$) than the mature, and the elder cattle, and this is due to body mass related to age which directly correlated to tsetse attraction [39].

Regarding cattle breed and the presence of *Trypanosoma* DNA (Fig 4D), all the above identified trypanosomes in cattle were found in the Arab zebu breed, as it was the largest group. The same observation was also in the White Fulani breed. Cattle of the M'bororo breed were only positive for *T. congolense* and *T. theileri*. Overall, *Trypanosoma* species DNA (Table 1) was in 34.2% (95% CI: 29.5–39.1%) of Arab zebu, 41.7% (95% CI: 25.5–59.6%) of White Fulani, and 28.6% (95% CI: 11.3% - 52.2%) of M'bororo group. And this difference was not statistically significant ($X^2 = 0.32$; $p = 0.56$).

## Packed cell volume (PCV) in relation to cattle breed, age and trypanosomes occurrence

The PCV value (%) of 370 animals was recorded in the field. Based on the recorded PCV and the questionnaire answered, 236 potentially sick animals had a mean PCV of 38.2±6.3 while 134 healthy animals averaged at 41.0±6.6, a statistically significant difference ($p < 0.0001$). Overall, correlating with poorer health status, the cattle from the refugee camp had the lowest

mean PCV (36.4±7.9) compared with the nomadic cattle (38.1±5.8) and the sedentary animals (41.1±7.1). Regarding the age, young cattle had the lowest PCV mean (37.1±6.8), while the mature cattle with a PCV mean of 40.4±6.5 and the elder with 38.8±6.1. There is no difference between the PCV mean of the M'bororo (40.8±7.0) compared to the White Fulani breeds (41.4 ±6.3), but the Arab zebu showed a slightly lower PCV mean (38.8±6.5). Within the Arab zebu breed, the distribution of the PCV average correlates to recorded health status (p<0.001).

*T. congolense*-infected cattle showed the lowest PCV mean (35.4±5.8) compared with *T. grayi*-infected (39.8±6.7), *T. vivax* (40.2±4.9) and *T. theileri*-infected cattle (clades taken together 40.4±6.4). Mixed infected cattle presented a PCV mean of 36.76±6.0, while all positive cattle taken together had 38.5±6.5 and the negative animals 40.0±6.7. Of 207 physically healthy cattle recorded in total, 112 were found with trypanosomal DNA including typical pathogenic species (mean PCV 39.0±5.6, n = 26), while the PCR-negative healthy cattle (41.3±0.7, n = 79) had the highest mean PCV. However, there is no statistically significant difference when comparing the PCV of PCR-positive healthy cattle (40.5±0.8, n = 55) with PCR-negative healthy cattle (41.3±0.7, n = 79).

## Tsetse flies survey data

During the first survey in February 2017, 50 traps were set in 8 spots in Mandoul and Maro, respectively. During this survey, only 20 tsetse flies were caught in Maro, and a single tsetse fly in the Mandoul focus (see S4 Table for details). Thereafter, the following surveys were focussed on the Maro area, where 156 additional tsetse flies were caught in 48 traps out of 117 additional traps set, with a highest mean catch of 0.47 tsetse/trap/day in December of 2018 (see S4 Table for details). Of the total of 177 tsetse flies, 98 (54.8%; 95% CI: 47.2–62.3%) were females and 79 (45.2%; 95% CI: 37.7–52.8%) males. *Glossina fuscipes* group was collected in Mandoul, while *Glossina fuscipes* and *Glossina tachinoides* were trapped in Maro.

During the surveys, the temperature was high (often above 40˚C) and the relative humidity low, the tsetse flies died quickly in the cages due to dehydration and other factors such as stress leading to unusually high mortality. Since these dead tsetse flies could not be dissected as planned, after removing proboscis, legs and wings DNA was extracted from the remaining bodies (Tsetse Remaining Bodies, TRB).

## Trypanosome identification in tsetse flies

DNA extract from 171 proboscises, 34 guts and 143 TRB were screened for trypanosomal ITS1. 34.5% (95% CI: 27.4–42.1%), 58.8% (95% CI: 40.7–75.8%), and 63.6% (95% CI: 55.2–71.5%) of proboscis, gut and TRB, respectively, contained trypanosomal DNA (S2A, S2B and S2C Fig for details). This frequency combined single and multiple occurrences (see S5 Table for details). *T. vivax*, *T. congolense*, *T. brucei ssp.*, *T. simiae*, *T. godfreyi*, and *T. grayi* a trypanosome identified in reptiles, were identified. One fly (TRB) showed DNA similar to *T. bennetti*, an avian trypanosome [40], termed *Trypanosoma sp.*-Maro1. The sequencing of its *gGAPDH* gene confirmed it. Trypanosomal DNA with some similarity to *Trypanosoma sp.* SDNK92 (ref. LC492122.1) was also identified in one gut and one proboscis from two different tsetse flies; termed *Trypanosoma sp.*-Maro2. The sequences similarities of both these unknown trypanosomes were less than 92% to the referred trypanosomes. Amplicons of about 150 bp and sequences similar to *Bodo caudatus* (203 bp) were also present but were not included in the trypanosomes' frequency. The single tsetse fly caught in the Mandoul focus was positive for *T. vivax*.

The overall trypanosome species distributions found within the positive proboscis, gut, and TRB are shown in Fig 5A and 5B and 5C, with *T. vivax* largely represented in all tissues.

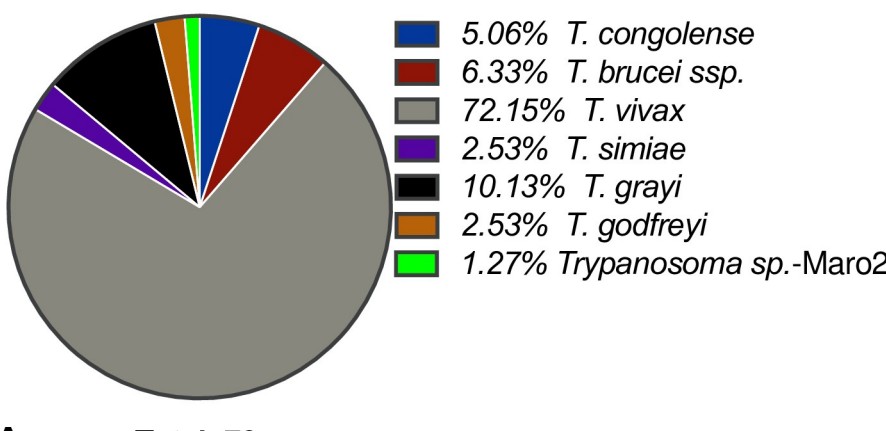

**A** Total=79

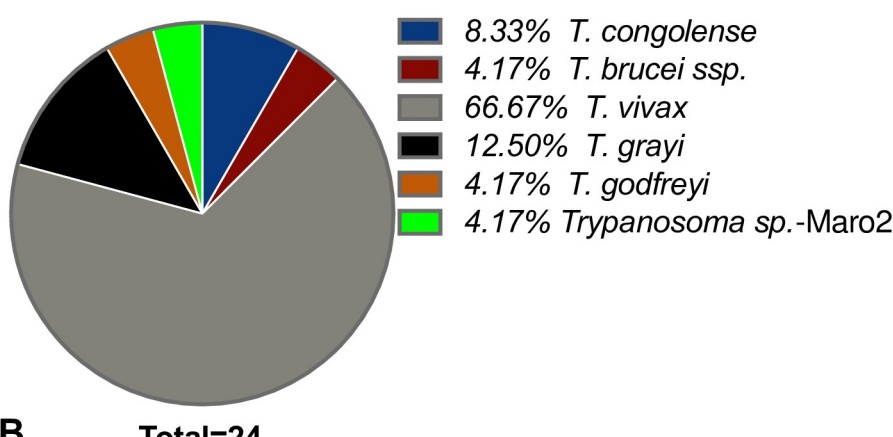

**B** Total=24

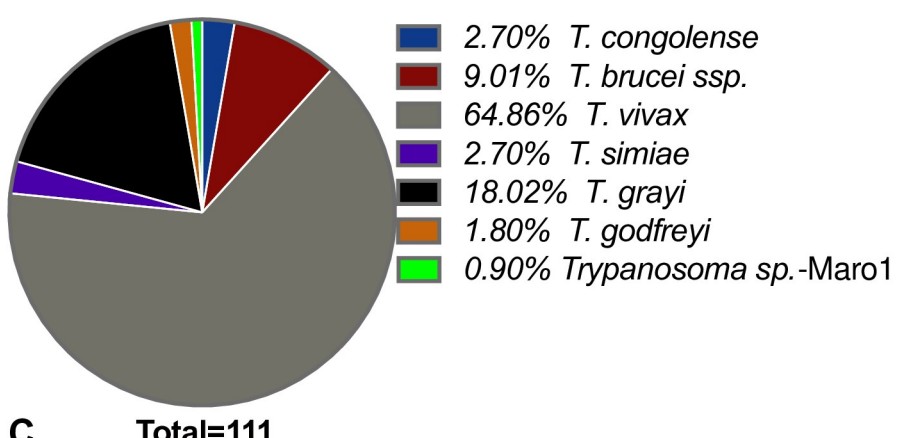

**C** Total=111

**Fig 5. The overall distribution of trypanosomes in positive tsetse fly samples. A**- Proboscis tissues; **B**- Gut tissues; **C**- Tsetse remaining bodies (TRB).

Remarkably, all typical livestock pathogenic species were detected in the tsetse fly vector. There was also evidence for DNA of *T. grayi* and *T. godfreyi* in the gut, the proboscis, and in the TRB (Fig 5A and 5B and 5C). Additionally, *T. simiae* was also detected, nevertheless only in the proboscises and the TRB.

At the Maro focus, 13% (95% CI: 8.3–18.7%) of tsetse flies were caught in the Canton Maro, while 86% (95% CI: 79.4–90.3%) were from Baguirgué site (Canton Gourourou) (see S4 Table for details). However, looking at the distribution of trypanosomes in these 2 locations at the same collection period (March) in the proboscis tissue, the species distribution was similar in most cases: *T. vivax* (50% of tsetse (n = 36) from Baguirgué and 52.6% of flies (n = 19) from Birya), *T. congolense* (2.8% in Baguirgué and 5.3% in Birya), similarly for *T. grayi* and *T. simiae*.

Mixed occurrences of trypanosome species DNA were observed in several samples (9.35% in proboscis, 11.76% in the gut and 16.78% in TRB). The most regular was the occurrence of *T. vivax* with one or two other parasites (S5 Table). In the gut tissue, *T. grayi* was identified with *T. vivax* and *T. brucei ssp*. In one proboscis, a presence of all cattle pathogenic species, *i.e. T. congolense*, *T. brucei ssp*., and *T. vivax* was observed.

## Summary of the most prominent findings

Overall, *T. b. gambiense*, the pathogenic species of humans and *T. congolense*, the pathogenic species of livestock were found in humans in the Maro sleeping sickness focus (Table 2). In contrast, in the Mandoul focus, only one person showed evidence for trypanosomal DNA. However, it belonged to an unknown *Trypanosoma sp.*-129-H. Regarding the sampled cattle, *T. vivax* was the most frequent trypanosome in Maro while *T. theileri* was found with very high frequency in Mandoul. *T. vivax* was the most frequent in tsetse flies in both foci including the single tsetse fly trapped in the Mandoul focus. Taken together, the results of this study showed evidence of a higher diversity of *Trypanosoma* species in the Maro area than in the Mandoul focus, including both human and livestock pathogenic species.

## Phylogenetic analysis of trypanosome species

*gGAPDH* sequences of trypanosomes circulating in the Mandoul and the Maro foci from representative human, cattle and tsetse samples were analysed for phylogenetic relationships and genetic diversity.

Three main clusters were observed when analysing 21 field samples and 8 reference sequences retrieved from GenBank database (Fig 6A). As expected, Salivaria trypanosomes formed one cluster. *T. congolense* and *T. b. gambiense* were closely related and formed a branch

**Table 2. Overview of trypanosomes frequency (in %) in humans, cattle and tsetse.** The lower and upper limits of the 95% confidence interval are indicated in parentheses.

| | Maro | | | Mandoul | | |
|---|---|---|---|---|---|---|
| | Nr. Collected | Trypanosome positive samples | Most frequent species | Nr. collected | Trypanosome positive samples | Most frequent species |
| **Human** | 392 | 3.3% (1.8–5.6) | *T. b. gambiense* *T. congolense* | 306 | 0.3% (0.0–1.8) | *Trypanosoma sp.*-129-H |
| **Cattle** | 462 | 34.4% (30.1–38.9) | *T. vivax* *T. congolense* | 78 | 82.1% (71.7–89.8) | *T. theileri* |
| **Tsetse** | 176 | | | 1 | | |
| **Proboscis** | 171 | 34.5% (27.4–42.1) | *T. vivax* | 1 | 100% | *T. vivax* |
| **Gut** | 34 | 58.8% (40.7–75.8) | *T. vivax* | 1 | 0% | |
| **TRB** | 143 | 63.6% (55.2–71.5) | *T. vivax* | 0 | 0% | |

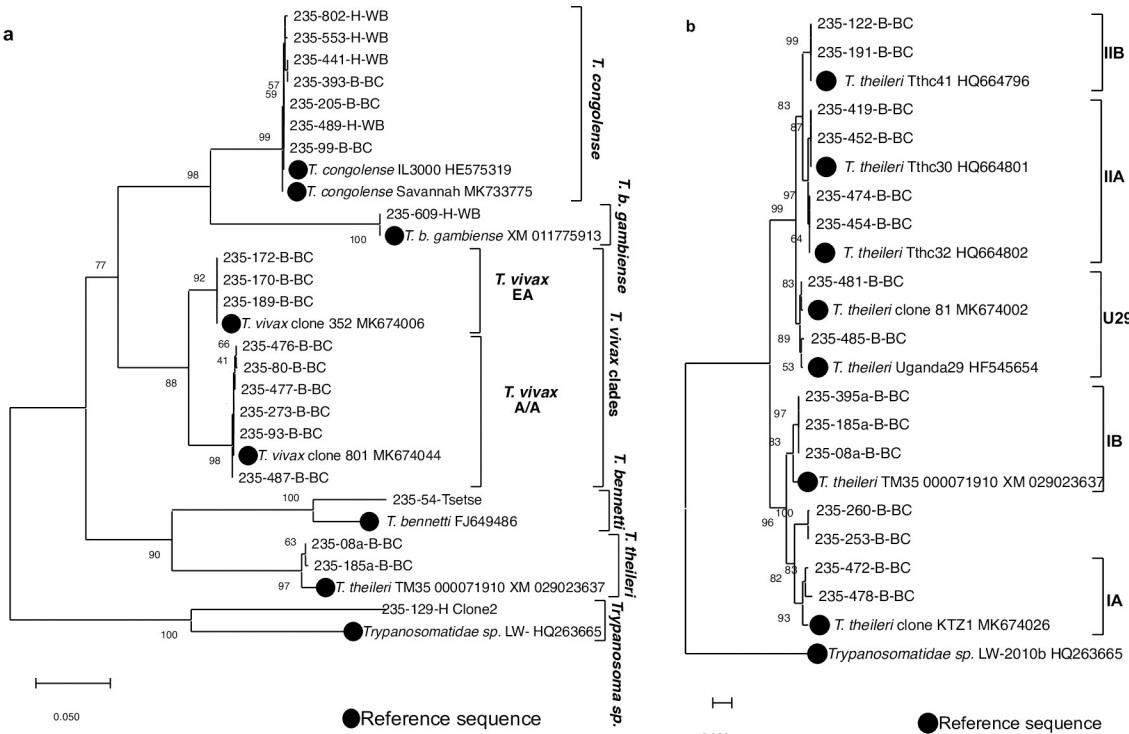

**Fig 6. Neighbour-Joining trees based on alignments of *gGAPDH* sequences from trypanosome species detected in human, cattle, and tsetse in Southern Chad.** They were calculated using complete gap deletion and tested with 700 bootstrap replications using MEGA X software (Kumar et al., 2018). **a-** *gGAPDH* nucleotide sequences of 21 representatives of different trypanosome species and 8 reference sequences retrieved from GenBank were aligned. **b-** *gGAPDH* nucleotide sequences of 15 representatives of *T. theileri* clades detected only in cattle samples and 8 reference sequences belonging to IA, IB, IIA, IIB and U29 *T. theileri* clades retrieved from GenBank were aligned. Evolutionary analyses involved 613 bp (**a**) and 563 bp (**b**) stretches. Abbreviations: EA, East Africa; A/A: Africa/America.

while *T. vivax* formed the second branch of this cluster. *T. vivax* showed 2 clades, the East African (*T. vivax* EA) and the African/American (*T. vivax* A/A, also called West African/South American type WA/SA). Similarly, known Stercoraria trypanosomes formed the second cluster including two branches: *T. theileri* and *T. bennetti*. The sequence of *Trypanosoma sp.*-Maro1 was closely related to *T. bennetti* (FJ649486, 92.6% similarity).

Interestingly, a third cluster was formed with the outgroup reference, and this concerned *Trypanosoma sp.*-129-H having 84.1% similarity to *Trypanosomatidae sp.* LW-2010b Dobs; HQ263665. This *Trypanosomatidae sp.* was previously found in a fly *Drosophila obscura*, (using trypanosome-*gGAPDH*) and *Hyena* (targeting trypanosomal ITS1).

*T. theileri* was widely distributed and very diverse (Fig 6B). Two main clades, clade IA and IB on the one side, and clade IIA and IIB on the other side were observed as previously described [28,38,41]. Interestingly, besides the sub-clades IIA and IIB, one other sub-clade was observed and thus the sequences were 99.8% similar to *T. theileri* sequences from the GenBank (references MK674002 and HF545654). Additionally, some sequences were closely related to clade IA; however, they formed a different sub-clade.

## Discussion

Different trypanosome species (See S1 Fig for details), including typical pathogenic and non-pathogenic species belonging to the Stercoraria and Salivaria trypanosomes, were identified

during this study. The evidence of high trypanosome frequency in tsetse flies (see S5 Table for details) vigorously supports our observation of high frequency in cattle and humans in Maro (Table 2). This suggests that tsetse flies are transmitting cattle and human infective trypanosomes in this focus, whereas in Mandoul, the low occurrence of tsetse flies correspond to a lower diversity and a different species pattern. It has to be kept in mind that our molecular approach does not confirm active parasite infections, as traces of DNA from a previous presence of parasites can be detected by this sensitive method. It should also be noted that the selection of animals presented for sampling by the herdsmen may have been biased due to their interest in presenting sick animals for possible treatment. And this was especially perceived when surveying the nomadic groups.

The noticeable relevant findings were the presence of *T. b. gambiense* DNA in humans in Maro, stressing the ongoing HAT infection risk of the area, besides unexpected *T. congolense* DNA in human samples which needs to be confirmed by direct microscopic observation of the parasite. In the Mandoul focus, there was no evidence for such species neither in cattle nor in humans. However, there was evidence of an unknown trypanosome, *Trypanosoma sp.*-129-H in one man cured of HAT, but with prevailing symptoms.

With regards to the overall parasites diversity, only very few *T. vivax* and *T. simiae*, and a high rate of *T. theileri* were detected in cattle in Mandoul. These species are known to be transmitted independently from the tsetse fly, which corresponds to the very low number of fly catches during our entomological survey. In Maro, tsetse flies were abundant, and HAT/AAT pathogenic species were present in a relatively high number of tsetse flies, in addition to previously undescribed trypanosomes, *Trypanosoma sp.*-Maro1 (*T. bennetti*-like) and *Trypanosoma sp.*-Maro2. The findings indicate differences between the two foci in terms of trypanosomes diversity potentially in relation to tsetse fly's abundance.

## Situation in humans

Regular screening campaigns of humans for HAT cases have been undertaken by the Ministry of public health and its partners within the historical HAT-foci in Southern Chad. The case is defined, if coagulation is present at a blood/lymph dilution of less than 1/8 when using Card Agglutination Test for *T. b. gambiense* (CATTs test), and to some extent LAMP assays and microscopy in the Mandoul focus. Whereas HAT cases have been decreasing over the years in the Mandoul focus [22,42], recently, the resurgence of 23 new cases was reported at the old Maro focus [43]. In this study, we identified *T. b. gambiense* in two human samples from the Maro area. It was detected in one child and one older man confirming the presence of the parasite in the area and the ongoing risk of HAT infections, as it was also identified in animal reservoirs reported by Vourchakbé *et al.*, 2020 [44]. These two participants have had no previous infection with this parasite. However, none of the samples collected in Mandoul was interestingly, *T. b. gambiense* positive, which could be due to the reduction of its incidence reported by Mallaye *et al.*, [21] and the low tsetse fly number. Thus, a wider monitoring is needed to state more precisely the overall parasite prevalence in the two foci.

An interesting observation was the presence of an older participant in the survey with a cured HAT infection. He still had the repercussions of HAT-like symptoms, when the survey was undertaken but had tested negative for *T. b. gambiense* in all the last active screening campaigns according to the health service of the locality. Unexpectedly, both PCR targeting *Kinetoplastida* ITS1 region and *gGAPDH*, evidence was obtained for the presence of an undescribed trypanosome. The sequences of *Trypanosoma sp.*-129-H DNA were related to unidentified *Trypanosomatida sp.* found in *Hyena* (using ITS-1, JN673399) and in *Drosophila obscura* (using *gGAPDH*, HQ263665). As the techniques used to diagnose HAT recommended by

WHO are very specific for *T. b. gambiense* [2] and sensitivity of microscopy analysis is limited, *e.g.* in case of low parasitemia, this parasite may have remained undetected for a long time. Thus, immediate further investigations would need to be undertaken to isolate and character-ise the parasite and furthermore look closely on the pathogenicity of this unknown trypano-some, in case of non-transient infection.

A second observation was the presence of *T. congolense* DNA in human blood samples in Maro. The abundant presence of *T. congolense* DNA we observed in cattle (Fig 3) indicated that also humans in this area are highly exposed to bites by tsetse flies transmitting *T. congolense* (which flies as well have been identified with its DNA traces) (Fig 5). This becomes evident, when looking at single settlements, for example, one of the nomadic communities, where one human, 28 cattle (16%) and 3 tsetse flies (13%) were *T. congolense* positive. Usually, humans are resistant to *T. congolense* due to innate protection, including most trypanosome species [45]. Among the innate protection, the trypanolytic factors (TLF1 and TLF2) [46] found in human serum are able to lyse trypanosomes upon entry [47], therefore serving as a natural host innate parasite defence mechanism. However, cases of atypical HAT have been reported [26,48], suggesting that *T. congolense* might be able to produce infections in man. Therefore, other investigations need to be undertaken on the infected humans to determine and discuss whether a pathogenicity and a zoo-notic potential could be revealed. This could involve initially an observation of the parasite using a direct microscopy. It is quite interesting that also other *Trypanosoma* species were circulating in the Maro area, however they were not detected in humans, as *e.g. T. vivax*.

## Situation in cattle

Apart from the direct risk of HAT for human health, AAT denotes a heavy burden on the live-stock and agriculture-based livelihoods of the people in rural areas. This implies, *e.g.*, the con-stant use of drugs such as trypanocides to treat the animals, as also noticed during these surveys; which is not without cost and contributes negatively to the economic development of the affected rural areas [2,49]. The animal-pathogenic species affecting cattle mainly include *T. congolense*, *T. vivax*, and, *T. b. brucei*.

As indicated above, the abundant presence of *T. congolense* in cattle in Maro (Fig 3) poses a high risk of AAT in this area. Regular surveillance is therefore needed to treat the respective animals. A promising finding was the absence of *T. congolense* in the Mandoul area, indicating that the tsetse control campaign effectively reduced *T. congolense*-induced AAT. However, due to the low number of cattle sampled in Mandoul, the obtained results cannot be extrapolated to the whole area.

The picture looks different for the presence of *T. vivax*. Though found in fewer of the tested cattle, *T. vivax* was detected in Mandoul as well as the Maro focus. This can be connected to the ability of *T. vivax* to be transmitted mechanically and stresses the need for additional con-trol measures, apart from tsetse fly control, when targeting this parasite.

Looking in more detail on the intraspecies diversity, *T. vivax* clustered in two clades. Both were present in samples collected from the Maro focus. One of these clades groups with the East African *T. vivax* (EA) with a strong homology and high similarity. This clade EA was described for the first time in Tanzania [50], then in Nigeria [29] and in Cameroon [38] in 2019, and now its presence in Chad was confirmed in this study. This leads to the suggestion that either this strain is spreading across the African continent (Central and Western Africa) or increased sequencing data are revealing a more detailed picture on the parasite diversity, which could not have been observed before, as also reported by Adams *et al.*, [51] on perfor-mance of molecular identification techniques. The second widespread clade, *T. vivax* A/A (African/American or WA/SA) [50], was present in samples from both foci.

*T. vivax* pathogenicity in cattle appears to be isolate-dependent. *T. vivax* A/A were described to be more pathogenic than EA isolates [52,10]. Furthermore, a strain-/subgroup-dependent pathogenicity level has been observed in the same geographical region depending on the infective species of the tsetse fly [51,52]. In agreement with the pathogenicity related to *T. vivax* subgroup, 3 cattle in the Mandoul focus died a few weeks after our survey, and the obtained sequences clustered with the *T. vivax* A/A clade. Already during sampling, these animals showed AAT-symptoms such as inappetence, asthenia, tearing, weight loss and oedema. This suggests that this strain might be responsible for AAT outbreaks in the area and possibly throughout the country, which might put the effort of fighting animal diseases under duress.

Several other trypanosomes were present in cattle samples, including *T. theileri*, a worldwide distributed parasite generally considered non-pathogenic. Analysis of *T. theileri gGAPDH* sequences revealed several sub-groups, among them the four known clades IA, IB, IIA, and IIB [38]. In this study, another clade (235-260-B-BC and 235-253-B-BC) closely related to clade IA was observed as well as a clade formed by the new strain found in Uganda (*T. theileri* Uganda29; HF545654) and in Cameroon (*T. theileri* clone 81; MK674002) [38]. This confirms that *T. theileri* is genetically diverse. Despite their presence in cattle, *T. theileri* was not observed in the tsetse samples which indicate its transmission to be independent from tsetse flies. It should be noted that this parasite was detected abundantly in Mandoul' cattle, where tsetse are close to elimination. Here, its frequency was much higher (82.1%; 95% CI: 71.7–89.8%), also higher than that obtained in previous studies; *e.g.* in Uganda (47%) [53] and in Northern Cameroon (30.5%) [38].

The genetic variation within *T. theileri* might lead to strain-dependent implications on the health status of the animals. Along this line, reports from Cameroon observed that cattle infected with clade IIB have lower PCVs than those infected with clade IA or IB [28,38]. In this study from Chad, cattle presenting *T. theileri* clade IA, have slightly low PCVs (34.33 ± 6.4) similar to those infected with *T. congolense* (35.48 ± 5.8), lower than those having *T. theileri* clade IIB (37.67 ± 8.5), *T. grayi* (39.8 ± 6.7) or cattle negative for trypanosomal DNA (40.03 ± 6.7). However, this outcome is not sufficient to speculate on the pathogenicity of these parasites (*T. theileri* and *T. grayi*) as many other parameters can change the PCV value. Nevertheless, already previous studies pointed out isolated cases of *T. theileri* pathogenicity [54,55], including a most recent cases in Italy [56]. These observations and the PCVs value linked to clades could give a path for research to determine whether *T. theileri* can be a pathogenic trypanosome under certain circumstances or when present in a certain genetic variant.

Further parasites were detected in cattle in Maro. Interestingly, evidence for *T. grayi* was obtained in 10 cattle blood samples. At first sight, this was unpredicted, since when describing this parasite, Hoare *et al.*, [57] could not infect mammals with this reptilian parasite. However, more recently, also in Cameroon *T. grayi* was detected in cattle [28,38]. Together, these observations support the note that there might be a burning issue of host adaptation of this trypanosome and change in its lifecycle. Thus, it is important to look more closely for *T. grayi* infections in cattle and its possible pathogenicity in this host.

## Tsetse fly data

Overall, a high diversity of trypanosomes was identified in tsetse, mirroring the diversity observed in humans and cattle in Maro.

The frequency of *T. vivax* found in tsetse flies was higher compared to other studies across the West and Central African tsetse fly area, where the highest was 34% overall positive [28], and 11.7% in all screened proboscises [29]. The presences in the gut tissues is likely to be remains of a recent blood meal since *T. vivax* is not expected to colonise the tsetse gut [58].

The sites of the development of trypanosomes during their life cycles in tsetse is species-specific [10,59]. However, already in previous studies, molecular identification of trypanosomes revealed unexpected sites of their DNA, for example, similar to our findings *T. grayi* in the proboscis [28,29] or *T. vivax* in the midgut [60]. This could be explained on the one hand by the sensitivity of the method, as it detects down to 10 pg DNA when field conditions were mimicked using tsetse fly midgut [27], and thus, during the transit of the parasites. On the other hand, it could be explained by residual DNA from a recent blood meal.

Described *T. bennetti* was initially found in the American Kestrel (*Flaco saoverius*) [61] and in European passerine birds and raptors confirmed by its isolation from nestlings and year-lings, suggesting its local transmission [62]. Similar sequences but not identical to this species was identified in one tsetse; *Trypanosoma sp.*-Maro 1. Interestingly, when performing blood meal analysis on this tsetse fly sample, a bird (*Ardea purpurea*) was revealed as its blood meal source. This is a piece of evidence that it is most likely to be a bird parasite.

As stated above, tsetse control activities were ongoing in Mandoul for the last 3 years and during our surveys. It is important to note that only one tsetse fly was caught in the Mandoul focus (which is a more confine area) during the surveys, although 20 traps were set at 8 different locations within 3 days. The rarity of the tsetse vector is most likely due to the success of tsetse control organised by IRED and its Partners by setting impregnated Tiny Targets that attract and kill tsetse. Also, Mahamat *et al.*, [22] observed a similar frequency with only 5 tsetse flies being caught during nine surveys at this area in 2017. Similar observations of the Tiny Targets effect on tsetse population reduction were reported in West and East Africa [63,64]. In contrast, in the Maro focus, the tsetse control has just started in 2018, after we collected most of the samples of this study. This probably played a role in the high trypanosome diversity observed in the area (see S1 Appendix, S3 and S4 Tables) as also more tsetse flies were found.

## Differences between the two areas and concluding remarks

Looking at the global picture of trypanosome distribution, a different pattern emerged in the two foci. Overall, Maro showed high diversity in tsetse-transmitted parasites. In Mandoul, diversity was much lower. On the one hand, this might be due to the reduced number of cattle samples tested in Mandoul. On the other hand, the parasites present in the blood of the surveyed animals shifted from animal pathogenic and tsetse transmitted parasites in Maro to generally considered non-pathogenic parasites (*T. theileri*) and parasites that do not only rely on tsetse flies for transmission in Mandoul (*T. vivax, T. simiae*). A higher rate of positive cattle accompanies this observation (Table 2). A similar pattern was observed by Paguem *et al.*, [38]. This is quite interesting, because it evokes some kind of competition: (1) Either the blood-sucking insects are competing and the reduction of one species, in this case, tsetse fly, benefits the growth of other blood-sucking flies such as Tabanids or *Stomoxys*. The presence of the *Tabanidae* and *Stomoxys* in the Mandoul area was previously reported [49]; however, their implication on the transmission of the parasites in the area has not been studied yet, which urge to be undertaken. (2) Or the trypanosomes are competing for the same hosts, and *T. theileri* might only be able to establish infections when the immune system is not activated against trypanosomes by pathogenic trypanosome species.

The identification of *Trypanosoma sp.*-129-H and *T. theileri* in Mandoul suggests that other trypanosome parasites are taking place and might be transmitted by other biting arthropods known as mechanical vectors.

It will be interesting to follow up, whether also in the Maro focus, the tsetse populations will also be reduced successfully, and see which impact this will have on the diversity of trypanosomes in this area. However, Maro is bordering the CAR and transhumance activities to and

from areas close to its National Park (Bamingui-Bangouran) in search of animal food supply, are frequent. This could impact in this region, as observed in the trypanosome's diversity that emerged from this study. The CAR savannah areas, known for their densely populated parts, are heavily infested with tsetse flies and potentially under the continuing threat of an AAT epidemic [21]. Unfortunately, due to the volatile and complex situation, no tsetse fly control is underway in the neighboring areas of the CAR, which could be involved in a joined effort to control the vector and the diseases. Thus, the Maro focus should get great attention, and more host species and individuals should be monitored.

Several other patterns could be observed in this study. The proportion of *T. vivax* at the beginning of the dry season (November) shifting to an increase towards the end of the dry season (March) (Fig 4B) is in agreement with a report from Nigeria [65], which stated its predominance in the dry season (and that of *T. congolense* in the wet season). This could be explained by the presence of other biting insects, throughout the season acting as mechanical vectors and maintaining the bovine trypanosomosis in the herd while tsetse populations are suppressed. Regarding trypanosomes distribution between cattle breeds, White Fulani group presented the highest frequency of *T. congolense*, *T. brucei ssp.*, and *T. godfreyi*, while *T. theileri* was widely predominant in M'bororo group and *T. vivax* in Arab zebu (Fig 4D). Our observation corroborates with that of Odeniran *et al.*, [65] who reported a high frequency of trypanosomes in White Fulani farms in Nigeria. This is due to the transhumance activity. Looking at this, results of nomadic animals included in this study (Fig 4A) reflect the observation with a high frequency of *T. vivax* followed by *T. congolense* and *T. brucei ssp*. This explains that transhumance activity would impact the transmission cycle and persistence of the parasites in the area [65], besides the susceptibility of the breeds [66] dependent on *Trypanosoma* species. However, for a conclusive picture, the seasonal impact of transhumance activities, and cattle breeds susceptibility need more longitudinal data addressing this issue.

## Conclusion

WHO had the goal to eliminate HAT as a public health problem by 2020, and the final goal of the sustainable disease elimination by 2030. In order to achieve this goal in Chad, the National Program with its partners have been organising campaigns; active screening and treatment of humans, as well as tsetse fly control by setting impregnated Tiny Targets. This strategy contributed to the reduction of tsetse populations and known pathogenic trypanosomes in the Mandoul area, as observed in the data emerging from this study. However, there is evidence for *T. theileri*, *T. vivax*, *T. simiae* in cattle and an unknown trypanosome in human which could lead to a resurgence and probable pathogenicity of the disease complex in the Mandoul area, and other vectors could play a role. Thus, the situation needs to be monitored. In contrast, the Maro area bordering the CAR could be an unknown reservoir of parasites. Based on its proximity with the CAR, which is having a complex and volatile social situation as well as the uncontrolled crossing of the borders of pastoralists and their livestock, it is highly expected that tsetse flies, others biting insects and various trypanosome species can move from one country to the other. As an outcome of this study, high diversity and frequency of trypanosomes have been observed in human, cattle, and tsetse fly vector including typical and atypical pathogenic species, suggesting a zoonotic potential leading to get close attention in this area. Therefore, to achieve the goal of eliminating HAT as a public health problem, all the players such as natural host and vector, and reservoirs and mechanical vectors have to be considered to exclude a resurgence of typical HAT from these sources and neighbouring areas, and atypical trypanosomiasis in case of non-transient infection.

## Supporting information

**S1 Appendix. Sequences accession numbers of trypanosome species identified in this study.**
(XLSX)

**S1 Fig. Gel picture showing the amplicon sizes of trypanosomal ITS-1.**
(TIF)

**S2 Fig. Percentage of positive and negative tsetse tissues for *Trypanosoma* sp. DNA.**
(TIF)

**S1 Text. Study areas description.**
(PDF)

**S2 Text. Samples collections and processing, and DNA extraction and quantification.**
(PDF)

**S3 Text. Molecular amplification and identification procedure, and subcloning and sequencing of amplicons.**
(PDF)

**S1 Table. Generic and specific primers used in the study.**
(PDF)

**S2 Table. Target villages for human blood samples collection.**
(PDF)

**S3 Table. Trypanosomes frequency in cattle and cattle sampled per village.**
(PDF)

**S4 Table. Tsetse collection sites, surveys duration, number of traps used and tsetse caught.**
(PDF)

**S5 Table. Trypanosomes frequency in tsetse fly tissues.**
(PDF)

**S1 Raw-Data. Human data.**
(XLSX)

**S2 Raw-Data. Cattle data.**
(XLSX)

**S3 Raw-Data. Tsetse fly data.**
(XLSX)

## Acknowledgments

We would like to thank the "Institut de Recherche en Élevage pour le Développement", and the "Programme National de Lutte contre la Trypanosomiase Humaine Africaine" (PNLTHA) in Chad for their collaboration and field survey supports at the beginning of this study, in particular Severin Mbainda, Brahim Guihini Molo, Richard Ouang and Nadjitessem Tanassingar as respectively nurse, entomologist, laboratory technician and veterinarian doctor and Peka Mallaye, the coordinator of PNLTHA. We are grateful to the administrative authorities and traditional leaders, especially the "Chef de Canton" of Bembaïtada, Maro, and Gourourou, without their support, samples could not have been collected. Special thanks go to Sister

Cecilia, Sister Titi, and their team for their services during our stay in the monastery. Special thanks are extended to AG Kelm members at the University of Bremen particularly to Frank Dietz, Mario Waespy, Archile Paguem, Jana Rosenau, Petra Seekamp and Nazila Isakovic for fruitful discussions and technical assistance and Sabine Limberg for the administrative support.

## Author Contributions

**Conceptualization:** Mahamat Alhadj Moussa Ibrahim, Sen Claudine Henriette Ngomtcho, Petra Berger, Hassane Mahamat Hassane, Sørge Kelm.

**Data curation:** Mahamat Alhadj Moussa Ibrahim, Petra Berger, Sørge Kelm.

**Formal analysis:** Mahamat Alhadj Moussa Ibrahim, Judith Sophie Weber.

**Funding acquisition:** Mahamat Alhadj Moussa Ibrahim, Sen Claudine Henriette Ngomtcho, Hassane Mahamat Hassane, Sørge Kelm.

**Investigation:** Mahamat Alhadj Moussa Ibrahim, Judith Sophie Weber, Sen Claudine Henriette Ngomtcho, Djoukzoumka Signaboubo, Petra Berger, Hassane Mahamat Hassane, Sørge Kelm.

**Methodology:** Mahamat Alhadj Moussa Ibrahim, Judith Sophie Weber, Sen Claudine Henriette Ngomtcho, Djoukzoumka Signaboubo, Petra Berger, Hassane Mahamat Hassane, Sørge Kelm.

**Project administration:** Mahamat Alhadj Moussa Ibrahim, Hassane Mahamat Hassane, Sørge Kelm.

**Resources:** Mahamat Alhadj Moussa Ibrahim, Judith Sophie Weber, Sen Claudine Henriette Ngomtcho, Petra Berger, Hassane Mahamat Hassane, Sørge Kelm.

**Software:** Mahamat Alhadj Moussa Ibrahim, Judith Sophie Weber.

**Supervision:** Hassane Mahamat Hassane, Sørge Kelm.

**Validation:** Mahamat Alhadj Moussa Ibrahim, Judith Sophie Weber, Petra Berger, Hassane Mahamat Hassane, Sørge Kelm.

**Visualization:** Mahamat Alhadj Moussa Ibrahim, Judith Sophie Weber, Sen Claudine Henriette Ngomtcho, Djoukzoumka Signaboubo, Petra Berger, Hassane Mahamat Hassane, Sørge Kelm.

**Writing – original draft:** Mahamat Alhadj Moussa Ibrahim.

**Writing – review & editing:** Mahamat Alhadj Moussa Ibrahim, Judith Sophie Weber, Sen Claudine Henriette Ngomtcho, Djoukzoumka Signaboubo, Petra Berger, Hassane Mahamat Hassane, Sørge Kelm.

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
