## [Decision Letter · Decision Letter 0]

27 Nov 2020

Dear Mr. Mahamat Alhadj Moussa,

Thank you very much for submitting your manuscript "Unexpected Trypanosoma species in humans, tsetse, and cattle, identified in Southern Chad: Mandoul and Maro sleeping sickness foci" for consideration at PLOS Neglected Tropical Diseases. As with all papers reviewed by the journal, your manuscript was reviewed by members of the editorial board and by several independent reviewers. In light of the reviews (below this email), we would like to invite the resubmission of a significantly-revised version that takes into account the reviewers' comments. 

Your manuscript has been reviewed by three experts in the field. All three agree that the study is interesting but all three agree that a number of significant weaknesses would need to be addressed before the manuscript could be accepted. Of particularly note are 1) the lack of clear articulation about the study goal and objectives, 2) improper use of statistics or failure to utilize statistical approaches to analyze the data, 3) lack of clear and detailed methods, 4) the need for additional rationale as to why the study size was chosen and 5) additional information is required on the ethical considerations. We will be happy to reconsider a revision that addresses these issues and the remaining reviewer comments detailed in the attached reviews.

We cannot make any decision about publication until we have seen the revised manuscript and your response to the reviewers' comments. Your revised manuscript will be sent to reviewers for further evaluation.

Sincerely,

Margaret A Phillips, Ph.D.

Deputy Editor

Margaret Phillips

Deputy Editor

Your manuscript has been reviewed by three experts in the field. All three agree that the study is interesting but all three agree that a number of significant weaknesses would need to be addressed before the manuscript could be accepted. Of particularly note are 1) the lack of clear articulation about the study goal and objectives, 2) improper use of statistics or failure to utilize statistical approaches to analyze the data, 3) lack of clear and detailed methods, 4) the need for additional rationale as to why the study size was chosen and 5) additional information is required on the ethical considerations. We will be happy to reconsider a revision that addresses these issues and the remaining reviewer comments detailed in the attached reviews.

Reviewer's Responses to Questions

**Key Review Criteria Required for Acceptance?**

**Methods**

-Are the objectives of the study clearly articulated with a clear testable hypothesis stated?

-Is the study design appropriate to address the stated objectives?

-Is the population clearly described and appropriate for the hypothesis being tested?

-Is the sample size sufficient to ensure adequate power to address the hypothesis being tested?

-Were correct statistical analysis used to support conclusions?

-Are there concerns about ethical or regulatory requirements being met?

Reviewer #1: A clear set of aims that link to each section of the Methods is not provided

There is no clear study design or description of statistical analysis used to inform the results

No power analysis provided

Reviewer #2: The methods needs to include details on the sample frame for both human and animal, random sample method is stated, but lacks details on either the number of samples that the survey set out to collect (cattle or human) or clear details on how the individuals were included for sample at each field site. 

In lines 432-435 of the discussion states that the owners presenting cattle for sampling [which is not random sampling]. Or are you suggesting that herders could have separated healthy animals from the herd to only take unwell individuals to the sample site?

The statistical evaluation that is referred to in the results section has not been identified.

Results section refers to a questionnaire (line 284), which is absent from the methods.

Cattle age forms part of the data analysis and discussion, how was the age of each animal established?

Reviewer #3: Sample size and selection

I tend to think that the methodology applied was appropriate for the study, although probably the objectives of the study should be stated more clearly. Particularly, I appreciate the sequencing, missing in other similar articles. On the other hand, sample sizes seem a bit low. Although clusters of villages, semi-nomadic, nomadic and military camps are mentioned, I’m not clear how participants (households and cattle) were selected/recruited: was there any randomisation that may allow to assume that the samples were representative? How comparable are the samples and the populations? Was there any sensitization campaign?

Ethical considerations

How were cases managed (i.e. human or cattle +ve samples? For example, were they and PNLTHA notified? Were +ve cattle treated, not able to recover…

Study areas:

It would be relevant to describe the main differences between the Mandoul and Maro foci/habitats. For example:

- Mandoul: Tsetse are restricted to the swaps formed in the southern limit of the Mandoul river. The tsetse distribution is limited in the South by the springs. As the river flows north, the swamp deteriorates into a marsh habitat, unsuitable for tsetse. Therefore, the population is isolated. Vector control operations with annual deployment of Tiny Targets started in 2014.

- Maro: In the southmost sections of Chari/Sido rivers, where rivers mark the border with CAR. Tsetse habitat configured by is the thin riverine vegetation along the banks of the rivers. Vector control operations started in the Chadian part in 2018, with annual deployments of Tiny Targets. No similar operations have been implemented across the border. 

Trypanosomes in tsetse:

Any reason not to analyse salivary glands for mature infections of T. brucei?

**Results**

-Does the analysis presented match the analysis plan?

-Are the results clearly and completely presented?

-Are the figures (Tables, Images) of sufficient quality for clarity?

Reviewer #1: Due to the absence of a discrete set of aims and the absence of details of study design (e.g. justification for selection of villages/ individuals), power analysis, or statistical analysis the results do not match the analysis plan. There are no estimates of uncertainty provided for trypanosome prevalence

Reviewer #2: Due to the two study areas being detailed I suggest the author presents each set of findings first as overall findings (2 areas combined) then Maro and Mandoul individually. Thereby standardising the order. To highlight the issue for a reader unfamiliar with the study; between line 174 and 177 it was not until I had already been confused by the reported percentages 0.5 and 2.7 (which I thought should have been 0.28 and 1.53) that it becomes clear the results are only Maro and not the whole study. 

Attention needs paying to typographical errors, to give some examples: Line 258 'at the tree time points' and line 341 where 'of' is missing after 13%. Further, throughout the results section the decimal places used in numbers is inconsistent.

There are sections in the results that belong in the discussion. examples; line 262-263 and 276-278.

Figure 1, the colour used for tsetse trap location is very hard to identify against the green base map, also the text in the image seems to be quite blurred.

Table 1 should be in the results section and referred to in the discussion.

Reviewer #3: Human cases

Two human cases of T. b. gambiense in Maro seems to suggest a relatively high prevalence. Do the authors consider that this may be a good representation of the prevalence in the foci? Or, may this relate to the low sample size and/or bias in the selection of participants? Or, ‘oversensitivity’ of the technique? I assume the main point of this finding is proving the presence of T. b. gambiense? Do the forms include a question about previous travels?

Tsetse:

Species not mentioned

Figures

Figures are small and low definition. They are difficult to read.

**Conclusions**

-Are the conclusions supported by the data presented?

-Are the limitations of analysis clearly described?

-Do the authors discuss how these data can be helpful to advance our understanding of the topic under study?

-Is public health relevance addressed?

Reviewer #1: It is not made clear why this study helps to better understand the situation in Chad. Without detailed methods and robust comparison between the two foci in the results, no conclusions can be drawn with respect to the differences between Mandoul and Maro.

Reviewer #2: line 679-280: Tiny targets are only used as tsetse control not elimination, also note tinny is a spelling error.

line 684-686: Maro is now receiving parasitological evaluation and regular control of tsetse (as you state in line 638).

The close of the conclusion is stretching. The assertion that to achieve the HAT elimination goal atypical trypanosomes need considering is at odds with the earlier statement in lines 399 - 403 where you highlight that the occurrence of non-HAT species of trypanosome detected in a human is likely the detection of an unviable infection.

Reviewer #3: 385-388 “To evaluate risk assessment of HAT, regular screening campaigns of humans for T. brucei gambiense using microscopy, Loop mediated isothermal amplification (LAMP) and RDT-kit have been undertaken by the Ministry of public health and its partners within the historical HAT-foci in Southern Chad” At the moment, case definition is Chad for HAT is based on CATTs test at certain titer dilution (I think 1/16). Other direct (e.g. LAMB, mAECT) or indirect (e.g. trypanolisis) tests are not as common as in other countries.

397-399 “It is the first study conducted in two Chadian HAT foci using a molecular identification tools, i.e. ITS1 amplification supported by gGAPDH analyses and sequencing to screen at the same time the tsetse fly vector, as well as human and cattle as definitive mammalian hosts”. I know there are differences, but please see https://www.parasite-journal.org/articles/parasite/full_html/2020/01/parasite200101/parasite200101.html

“Diversity and distribution of trypanosomes in the area”:

431-432 “The evidence of high trypanosome frequency in cattle (see S5 Table for details) vigorously supports our observation of high frequency in tsetse flies” In my view, the data shared by the authors show a clear distinction between the findings between Mandoul and Maro. Thus:

Mandoul:

- Hard to find flies (because probably there are not many)

- T. theileri is predominant in cattle (transmitted mostly by tabanids), with some T. vivax (which can be transmitted by other biting insects)

- Neither T. congoloense nor T. brucei found (transmitted by tsetse)

- Do the above imply that the incidence of tsetse-transmitted tryps might be approaching zero? 

Maro:

- Tsetse were collected in relatively good numbers

- T. congolense predominant

- More variability in Tryps spp

- Do the above suggest that in Maro tsetse are vectoring human and cattle Tryps?

Could the vectors explain the differences in the seasonality for each Tryp spp?

The authors mention in the discussion that cattle participating in the study might have been biased, as perhaps owners presented sick animals hopping for a treatment. This might explain a relatively high infection rate in cattle. Was there any bias in the selection of human participants? (see above)

440-442 “The relevant result emerging from the data is the high frequency (Table 1) and diversity (see S5 and S7 Tables for details) of trypanosomes in cattle in both foci” As mentioned above, I don’t see it that way. I think the most relevant result is precisely the differences between both sites.

442-444 “Another finding and not the least was that, in the Mandoul focus T. theileri was abundant, whereas only very few T. vivax and T. simiae were found and there was no evidence for T. congolense, neither in cattle nor in humans”. This seems a repetition of the results without the discussion. How do the author explain this result? See above

“Diversity of known trypanosomes”

446-450 “The T. b. gambiense DNA that was detected in one child and one old man in the Maro focus, confirmed the presence of the parasite in the area…”. The authors state that the two cases proved the presence of T. b. gambiense in Maro. Instead of “Interestingly, none of the samples collected in Mandoul was T. b. gambiense positive, and this is in agreement with the reduction of its incidence reported by Mallaye” the author should be consistent with the way reporting for Maro, i.e. confirmed presence vs. non-confirmed presence. Is this “in agreement with the reduction in the incidence”? Probably yes, but this is not proof of it. With this small sample size, rather than using the data to infer an epidemiological situation, I think it would be more honest to use the results in terms of proven or non-proven presence of the parasite.

456-459 “Interestingly, T. vivax clustered in two clades, with both present in samples collected from the Maro focus. One of these clades groups with the East African T. vivax (EA) with a strong homology and high similarity. This clade EA was described for the first time in Tanzania [49], then in Nigeria [28] and in Cameroon [31] in 2019, and now in Chad in this study”. If the clade has been identified in Nigeria and Cameroon, why is it so surprising finding it in Chad? Any speculation about how the dispersion and diversity of T. vivax across Africa?

476-477 “T. theileri was not observed in the tsetse samples which confirms its transmission independently from tsetse.” The sentence if quite vague. We know that T. theileri is mostly transmitted by tabanids. In my opinion, the main point to be made about T. theileri in Maro and Mandoul is that the predominant Tryps spp in Mandoul is not transmitted by tsetse (because tsetse are close to annihilation); on the other hand, Tryps spp in Maro are more diverse.

487-489 “The identification of T. godfreyi and T. simiae in different tsetse fly tissues and cattle, common across the African continent [56], [57] suggests that the vector would have fed on wild animals such as warthogs” Why?? The population of domestic pigs are far larger than that of warthogs. Presumably, T. simiae are a problem for pigs.

513-514 “The more the animals get old, the more their susceptibility to pathogenic species is high, as age-related resistance to trypanosomes is recognised” As explained by Vale, Torr and others, the body mass of the host is directly correlated to the attraction of tsetse; thus, old (and heavy) cattle attract more tsetse than calves.

Relevance for the area(s)

624-629 If by the “elimination campaign” the authors refer to the use of SIT, as far as I know, this has not started yet. No release of sterile males was done during the time reported in this manuscript. The elimination project (using SIT) was attached to an existing project aiming to eliminate HAT transmission (not necessarily tsetse). And yes, that is being done using Tiny Targets, although PATTEC has nothing to do with it.

638-639 There isn’t any tsetse elimination campaign in Maro. There is a tsetse control operation, aiming (in addition to case detection and control) to eliminate the transmission of HAT. It started in 2018. Note than the river is the natural border between Chad and CAR (sometimes the river is just 5-10 m wide, some time a few hundreds) and all the efforts were done on the Chadian side, only.

Conclusions:

677 It might be worth mentioning somewhere that the current WHO goal is the “elimination of HAT transmission” by 2030.

679 As above, the aim of the Tiny Targets is not the elimination of tsetse. In Mandoul, and Mandoul only, there is a new project to implement SIT, but (if I’m not wrong) the release of sterile males has not started yet.

**Editorial and Data Presentation Modifications?**

Reviewer #1: (No Response)

Reviewer #2: Attention needs paying as references are out of order, eg: line 629-630 references Mahamat et al [23] but in the references on line 777 m[23] is Targeting Tsetse on LSTM's website.

Reviewer #3: 68-69 “Despite the WHO goal to eliminate it by 2020, HAT is still a public health problem, because 70 million people in 36 sub-Saharan African countries are at risk of infection”. Actually, the 2020 goals established by WHO were globally achieved by 2018; that was according to their own definition of “public health problem” (i.e., <1/10,000 cases, >90% of endemic foci; and <2,000 cases worldwide). The current goal is the elimination of transmission.

87-89 “The main insect vector in Africa are flies of the genus Glossina (Glossinidae: Diptera). However, the parasites can also be transmitted mechanically by other biting flies such as tabanids and Stomoxys”. Although, this is true for some Trypanosoma spp (e.g. evansi, theileri, vivax…), I find the statement a bit too general.

Figures are small and low definition

**Summary and General Comments**

Reviewer #1: Please see attached document

Reviewer #2: This represents interesting and insightful work, which I want to see published. 

There are aspects of the paper that need refining which will greatly benefit the reader. 

Details of some methods that have been used are absent from the appropriate section.

Results section will benefit from being revised.

The assertion that atypical trypanosomes could threaten HAT elimination should be accompanied by a reminder of your sentiment in lines 399-403 thereby ensuring the reader takes away an unbiased conclusion. 

From my knowledge the fieldwork was undertaken for this project predates work on eradicating tsetse in Mandoul, there was ongoing tsetse control by the consortium of PNLTHA, FIND, IRD and LSTM as part of the drive against gHAT. As such reference to eradication in Mandoul is misleading and should be reworded as ongoing tsetse control.

The article repeatedly refers to the FIND, IRD, LSTM operation as tsetse elimination, which is inaccurate, the project is objective to control tsetse to thereby reduce the risk of infection in gHAT foci and not to eliminate tsetse, this should be addressed throughout.

Abstract refers to trypanosomosis in humans, this should be trypanosomiasis for disease in humans.

Reviewer #3: The authors present in the manuscript a comprehensive study of the two main HAT foci in Chad, including parasitology, epidemiology, entomology, population genetics… The amplitude of the study has advantages and disadvantages. On the one hand, it is very informative (the authors report the Tryps spp circulating in both foci), but on the other hand, it lacks an obvious ‘selling point’ that may appeal potential readers. After reading the manuscript with interest, in my opinion, the data contains some implications that are not properly highlighted and discussed, and other perhaps less important with too much detail. The weakest part is probably the discussion, when the authors have the chance to convince readers why their work is relevant. It is indeed useful to know the Tryps spp circulating in humans, cattle and tsetse vectors, but in practical terms, some spp are more important than others.

For example, the authors dedicate extensive paragraphs to discuss things like ‘atypical HAT’, or Tryps spp than hardly cause any symptoms in livestock; spp of less medical or veterinary importance can be mentioned briefly, and expand on, for example, T. b. gambiense, T. b. brucei, T. congolense, T. vivax and perhaps, T. simiae. Conversely, what it is probably the most interesting points of the research are not discussed in the length they deserved.

My first advise for the authors would be to recognise the weakest points in the data, and highlight the strongest ones. For example, the sample size seems low, and probably the recruitment of cattle and people were not randomised. That means, the %s reported may not represent the whole population (not valid for epidemiological extrapolations). That is not a big problem, as they can still report relative frequency of the different parasites (in flies, cattle and human), and for those less common, presence confirmed vs. presence non-confirmed.

As a reader, I would like to see the differences between both foci: what those differences are and why. To show this, the reader needs to understand from the beginning what makes Mandoul and Maro so different:

- As a tsetse habitat, Mandoul represents an ecological island: it is isolated, and the risk of reinvasion is very low.

- Before 2014, when the only approach to control HAT relied solely on case detection and treatment, Mandoul was by far the most active HAT focus in Chad (some numbers would be good). The addition of vector control reduced dramatically the number of cases to a handful.

- As the number of cases in Mandoul declined, PNLTHA increased the efforts to identify new cases in areas that up to that moment were considered of less importance (including Maro). I’m quite convinced that the increase in the number of cases in Maro was the result of the additional efforts (at least in part).

- The tsetse habitat in Maro, on the other hand, is open for reinvasions. The deployment of targets started in 2018 and probably did not have any impact in this study. In the long term and due to the differences, we don’t expect to have the same results as in Maro.

With this background, the data suggest that the presence of Tryps in Mandoul can be explained without tsetse, but the opposite is true in Maro, whereas in Maro tsetse-transmitted Tryps were found in larger proportions, and the spp variability was much greater. The difficulties to collect tsetse in Mandoul seems to support this idea. Tsetse-transmited Tryps are of medical or veterinary importance (e.g. T. b. gambiense, T. congolense…), so that looks like a good thing. This could lead into a subsection in the discussion: practical implications; or what we need to do in Maro, so it can look more like Mandoul.

This contrasts with the recently published paper by Vourchakbe et al

https://www.parasite-journal.org/articles/parasite/full_html/2020/01/parasite200101/parasite200101.html

Vourchakbe and colleagues reported a higher proportion of T. b. g. in Mandoul, compared to that of Maro (Tryps found in goats, sheep, dogs and pigs, and they did not sequence the amplicons).

Probably there isn’t enough data to support big statements about potential trypanotolerant cattle breeds. We also need to consider other variables associated with breeds: for example, Arab zebu might look like trypanosensitive according to infection rates, but as the breed of nomadic heads, they are also exposed to different risks. Cattle are specially exposed when they cross rivers, and nomadic cattle does that more often than stable cattle. Another factor to consider (commented briefly by the authors) is the potential correlation between seasonality of certain Tryps spp and their vectors (e.g. tsetse, horseflies…). And, talking about vectors, the authors should mention the name of the tsetse, at least once: Glossina fuscipes fuscipes.

If the authors agree with me, they should probably indicate more clearly in the introduction the objectives of the study (e.g. to compare presence/absence of different Tryps spp -- in cattle, people and flies -- in Mandoul and Maro, or something along those lines). Then, the methods will fit with the objectives easily.

In agreement with that, I would also recommend a change in the title. First, I’m not sure why the current title is “Unexpected Trypanosoma species…”. I think the species found fit reasonably well with what it was expected. Furthermore, to me, the title should highlight the differences found in the two areas. This is to me the main ‘’selling point” of the article.

For all this, I’ll recommend the editor to accept the manuscript, although I think it should be reviewed by the authors, specially the discussion section.

PLOS authors have the option to publish the peer review history of their article (what does this mean?). If published, this will include your full peer review and any attached files.

Reviewer #1: No

Reviewer #2: No

Reviewer #3: Yes: Inaki Tirados
---

## [Decision Letter · Decision Letter 1]

24 Feb 2021

Dear Mr. Mahamat Alhadj Moussa,

Thank you very much for submitting your manuscript "Diversity of trypanosomes in humans and cattle in the HAT foci Mandoul and Maro, Southern Chad - A matter of concern for zoonotic potential?" for consideration at PLOS Neglected Tropical Diseases. As with all papers reviewed by the journal, your manuscript was reviewed by members of the editorial board and by several independent reviewers. The reviewers appreciated the attention to an important topic and overall were happy with the modifications you made in revision. A few minor revisions are requested at the stage. I will be able to accept the paper once you address these remaining issues.

Sincerely,

Margaret A Phillips, Ph.D.

Deputy Editor

Margaret Phillips

Deputy Editor

Reviewer's Responses to Questions

**Key Review Criteria Required for Acceptance?**

**Methods**

-Are the objectives of the study clearly articulated with a clear testable hypothesis stated?

-Is the study design appropriate to address the stated objectives?

-Is the population clearly described and appropriate for the hypothesis being tested?

-Is the sample size sufficient to ensure adequate power to address the hypothesis being tested?

-Were correct statistical analysis used to support conclusions?

-Are there concerns about ethical or regulatory requirements being met?

Reviewer #2: Methods are clear and well written.

Study design is well set out and clear, with clarification on the overall population in the study area

Ethical regulations are well set out as are the steps surrounding the participant's consent.

There are a few editorial issues that should be crosschecked and cleared up - line 220 to 245 appear to have a different spacing in the PDF.

Some of the terminology could be improved upon to be clearer I have included examples in the PDF.

PCV should be detailed in the relevant 'blood collection' section of the methods.

Reviewer #3: Comments and suggestions from the first review addressed adequately

**Results**

-Does the analysis presented match the analysis plan?

-Are the results clearly and completely presented?

-Are the figures (Tables, Images) of sufficient quality for clarity?

Reviewer #2: Analysis is described in methods with clearly presented results.

There are some matters that I have tried to highlight as best I could in the PDF attached.

The inclusion of the military camp was requested and should be reported in the findings but, as it was not part of the study design these samples should be separated from the study population in the results and analysis.

Reviewer #3: Comments and suggestions from the first review addressed adequately

**Conclusions**

-Are the conclusions supported by the data presented?

-Are the limitations of analysis clearly described?

-Do the authors discuss how these data can be helpful to advance our understanding of the topic under study?

-Is public health relevance addressed?

Reviewer #2: Conclusions are well thought out and supported. with presentation of the public health importance and highlighting specific findings of this study that do merit follow up investigation - both in human and animal health.

I am cautious about the presentation of the detection of T. congolense DNA in a human sample. The DNA presence is likely to be due to an unsuccessful infection that the individual has cleared. To be classified as an atypical-human trypanosome parasitological confirmation is required. There is a very strong case for follow up on the individual, but until that is done the issue is a peculiarity and not 'alarming' as described in line 564.

The treatment of animals with trypanocides is an interesting point that might add clarity to the situation with AAT. If the questionnaire (described in the methods) did capture any data on trypanocide usage it would be a strong addition, even 'unstructured interview' evidence of treatment or not that may have been given by cattle owners would be an interesting addition.

I would really like to know if the 2 people that were detected by PCR as having evidence of gHAT have been followed up - or are planned to be followed up, and the conclusions would be a nice place to have that.

Reviewer #3: Comments and suggestions from the first review addressed adequately

**Editorial and Data Presentation Modifications?**

Reviewer #2: very minor revisions to the body of text

Reviewer #3: Some teeny-weeny comments:

103-104. “However, AAT has remained a major obstacle to its development, which employs more than 40% of the population”. I don´t understand the sentence: does AAT employ 40% of the people in Chad?

104-105. “Chad also faces the public health problem HAT” I understand the sentence, although it sounds a bit odd.

118 “Animals may harbour the human pathogenic species, serving as a reservoir”. Although there are some studies suggesting a significant role of animal reservoirs in the transmission of g-HAT, I don’t think this is 100% conclusive, e.g. see Rock et al 2017

615-616. “Sequence analysis of T. theileri gGAPDH sequences revealed several sub-groups…” Remove first “Sequence”

688-689 “...however, their implication on the transmission of the parasites in the area was not studied yet”. Consider using the perfect tense (i.e. ‘has not been studied’), instead of the simple past (‘was’)

**Summary and General Comments**

Reviewer #2: This is a very good presentation of your work, which will benefit from some very minor adjustments as detailed.

Reviewer #3: I am glad to see how the authors have addressed and incorporate most of the suggestions from the first review. In my eyes, the current version of the manuscript makes an interesting reading, discussing between the findings in Maro and those in Mandoul (vis-à-vis the differences in the habitat, tsetse control operations, etc.). This offers an informative and stimulating article for people interested in trypanosomiasis control. Despite being relatively close, Maro and Mandoul illustrate two very different situations of HAT and AAT transmission; this is properly discussed in the ms.

PLOS authors have the option to publish the peer review history of their article (what does this mean?). If published, this will include your full peer review and any attached files.

Reviewer #2: No

Reviewer #3: Yes: Inaki Tirados

Figure Files:

Data Requirements:

Reproducibility:

References

---

## [Editor Report · Decision Letter 2]

23 Mar 2021

Dear Prof. Kelm,

We are pleased to inform you that your manuscript 'Diversity of trypanosomes in humans and cattle in the HAT foci Mandoul and Maro, Southern Chad - A matter of concern for zoonotic potential?' has been provisionally accepted for publication in PLOS Neglected Tropical Diseases.

Best regards,

Margaret A Phillips, Ph.D.

Deputy Editor

Margaret Phillips

Deputy Editor

---

## [Editor Report · Acceptance letter]

4 May 2021

Dear Prof. Kelm,

We are delighted to inform you that your manuscript, "Diversity of trypanosomes in humans and cattle in the HAT foci Mandoul and Maro, Southern Chad - A matter of concern for zoonotic potential?," has been formally accepted for publication in PLOS Neglected Tropical Diseases.

Best regards,

Shaden Kamhawi

co-Editor-in-Chief

Paul Brindley

co-Editor-in-Chief
